# Room-temperature polariton condensate in a quasi-2D hybrid perovskite

Marti Struve [1], Christoph Bennenhei [1], Hamid Pashaei Adl[1], Kok Wee Song [2], Hangyong Shan [1], Nadiya Matukhno[1], Jens-Christian Drawer [1], Sven Stephan[3], Falk Eilenberger [4,5], Naga Prathibha Jasti [6,7], David Cahen [7], Oleksandr Kyriienko [8], Christian Schneider [1] & Martin Esmann [1] ✉

Quasi-2D halide perovskites are chemically synthesized realizations of quantum well stacks with giant exciton oscillator strengths, tunable emission spectra, and very large exciton binding energies. While these features render quasi-2D halide perovskites a promising platform for room-temperature polaritonics, bosonic condensation and polariton lasing in quasi-2D perovskites have so far remained elusive at ambient conditions. Here, we demonstrate room-temperature cavity exciton-polariton condensation in mechanically exfoliated crystals of the quasi-2D Ruddlesden-Popper iodide perovskite $(BA)_2(MA)_2Pb_3I_{10}$ in an open optical microcavity. We observe a polariton condensation threshold of 0.41 μJ cm$^{-2}$ per pulse and detect a strong non-linear response. Interferometric measurements confirm the spontaneous emergence of spatial coherence across the condensate with an associated first-order autocorrelation reaching 0.6 with 1 ps coherence time and an effective de Broglie wavelength of 13 μm. Our results lay the foundation for a new class of room-temperature polariton lasers based on quasi-2D halide perovskites with great potential for hetero-integration with other van-der-Waals materials and combination with photonic crystals or waveguides.

Hybrid organic-inorganic halide perovskites (HaPs) have recently gained considerable attention driven by fast-paced progress in photonic, optoelectronic, and photovoltaic applications[1–4]. A particularly intriguing class of these hybrid materials is layered quasi-2D HaPs, chemically synthesized realizations of multi-quantum well (QW) stacks, represented by inorganic layers sandwiched between organic spacers (see Fig. 1a, zoom-in)[5–10]. Since the spacers act as potential barriers for electronic excitations in the HaP stack, low-energy electronic excitations are well confined in the inorganic parts. Quasi-2D HaPs thus share strong structural similarities with conventional multi-QW stacks, but feature significantly larger exciton binding energies. In

fact, excitonic correlations are so prominent in quasi-2D HaPs that almost all of their optical features are defined by exciton physics even at room temperature[11–15]. Just like in multi-QW stacks, the number of layers can scale up the overall exciton oscillator strength in quasi-2D HaP crystals, which is a distinct advantage over transition metal dichalcogenides (TMDCs), where tedious mechanical assembly is required for multi-layers. Unlike for their 3D counterparts, which have recently emerged as a promising material platform for monolithic room-temperature polariton devices[16–20], choosing the thickness of the inorganic layers in terms of the number of unit cells, $n$, results in a unique control over the exciton resonances in quasi-2D HaPs[7,21–23]. Even

[1]Institut für Physik, Fakultät V, Carl von Ossietzky Universität Oldenburg, Oldenburg, Germany. [2]Department of Physics, Xiamen University Malaysia, Sepang, Malaysia. [3]University of Applied Sciences Emden/Leer, Emden, Germany. [4]Fraunhofer-Institute for Applied Optics and Precision Engineering IOF, Jena, Germany. [5]Institute of Applied Physics, Abbe Center of Photonics, Friedrich Schiller University, Jena, Germany. [6]Department of Chemistry, Bar-Ilan Univ, Ramat Gan, Israel. [7]Department of Molecular Chemistry and Materials Science, Weizmann Institute of Science, Rehovot, Israel. [8]School of Mathematical and Physical Sciences, University of Sheffield, Sheffield, UK. ✉e-mail: m.esmann@uni-oldenburg.de

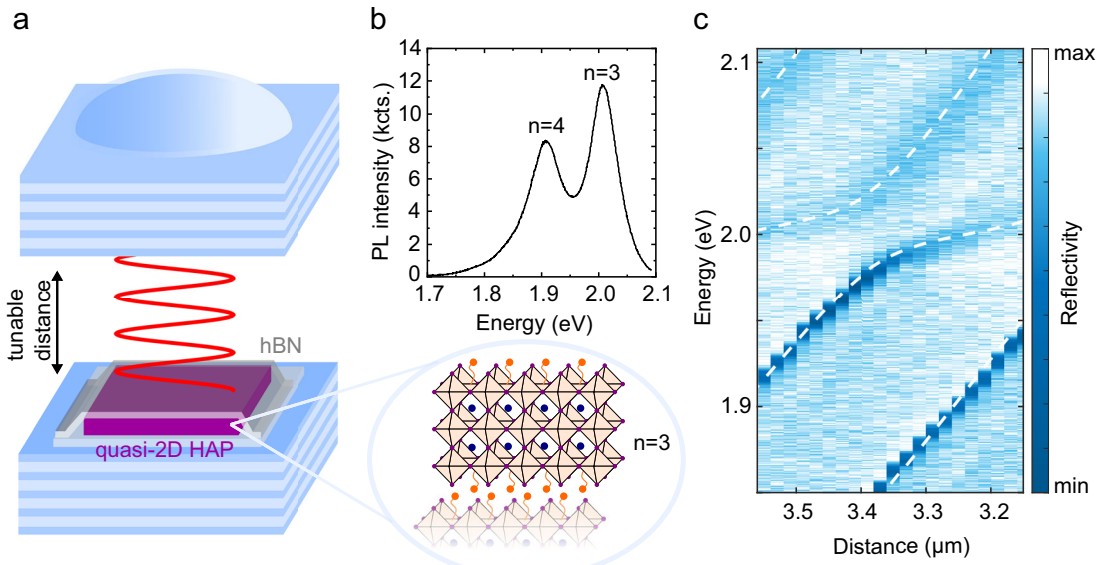

**Fig. 1 | Open cavity hosting quasi-2D perovskite and strong coupling condition. a** Two distributed Bragg reflectors (DBRs) on separate piezo stages form an open optical microcavity with tunable resonance. The upper mirror contains planar sections and sphere cap shaped indentations. The active material is a mechanically exfoliated, fully hBN-encapsulated quasi-2D halide perovskite (HaP) flake. **b** (top) Photoluminescence (PL) spectrum of a 218 nm thick $n = 3$ quasi-2D HaP crystal excited at 532 nm (CW laser) containing some admixture from $n = 4$ layers. (bottom) Structure of the quasi-2D HaP with $n = 3$. **c** White light reflectivity spectra as a function of the cavity air gap at normal incidence ($k_\parallel = 0$). From a coupled oscillator model (dashed lines) we deduce a light-matter coupling strength of $g = 23.5$ meV for the $n = 3$ quasi-2D HaP exciton at 2 eV.

the exciton binding energy is a separate, directly accessible tuning parameter in quasi-2D HaPs, e.g. via doping the organic spacer[24]. Most importantly, 2D HaPs can be micromechanically cleaved and re-assembled in a well-defined manner[4,25–28]. This has already lead to first demonstrations of charge-tunable gated devices in 2D HaPs[29] with electrically controlled optical response.

The giant excitonic oscillator strengths and binding energies, alongside with spectral tunability and versatility for complex hetero-integration render quasi-2D HaPs a particularly promising material class for exploring room-temperature polaritonics. The formation of cavity exciton-polaritons based on these materials has recently been observed in planar dielectric, metallic and hybrid cavities[30–37]. However, in quasi-2D HaPs, the bosonic condensation of exciton-polaritons[38,39] − their most significant feature for applications based on their quantum coherence − has only been demonstrated at 4 Kelvin in the 2D HaP (PEA)$_2$PbI$_4$ ($n = 1$, PEA=phenethylammonium)[40], and so far remained elusive at room temperature.

Here, we demonstrate room-temperature cavity exciton-polariton condensation based on mechanically exfoliated crystals of the quasi-2D Ruddlesden-Popper perovskite (RPP) (BA)$_2$(MA)$_{n-1}$Pb$_n$I$_{3n+1}$ ($n = 3$, BA=butylammonium) integrated into an open optical microcavity[36,37,41–44]. Under strong, non-resonant optical pumping, we observe a polariton condensation threshold at $P_{th} = 0.41 \mu$Jcm$^{-2}$ per pulse. We detect a significant blueshift of 0.84 meV in the condensed phase, which we attribute to non-linear processes based on Coulomb interactions between excitons as well as phase space filling effects. Through interferometric measurements, we confirm the emergence of spatial coherence across the condensate with a value of the first-order autocorrelation reaching up to 0.6 with an associated coherence time of 1 ps and an effective de Broglie wavelength of up to 13 μm. Our results lay the foundation for a new class of room-temperature polariton lasers based on quasi-2D perovskites with great potential for hetero-integration with other van-der-Waals materials for charge-tunable devices, electrically driven room-temperature polariton lasers, and for the combination with complex photonic crystals.

## Results

Figure 1a illustrates the implementation of our sample, based on an open optical cavity setup[41]: Two distributed Bragg reflectors (DBRs) made from alternating quarter-wave-layers of SiO$_2$ and TiO$_2$ are mounted on two separate xyz piezo stacks, forming an air-gapped Fabry-Perot resonator. In contrast to monolithic approaches, this configuration remains fully flexible in terms of cavity resonance by tuning the mirror distance and relative positioning in-plane. The top DBR contains both planar sections and sphere cap-shaped indentations (6 μm diameter, 150 nm depth), resulting in planar Fabry-Perot and three-dimensionally confined, Laguerre-Gaussian type[45] optical modes, respectively. We mechanically exfoliated 200-300 nm thick ~50 μm x 50 μm sized flakes of chemically synthesized quasi-2D HaPs and transferred them onto the bottom mirror by dry-gel stamping[46]. Since we operate at ambient conditions, we fully encapsulated the flakes with ~10 nm thick hexagonal Boron Nitride (hBN) layers eliminating both photo-oxidation and absorption of water[27] (see Methods section and Supplementary Section S1 for details). The flakes that were selected for the subsequent polariton studies were $n = 3$ quasi-2D HaPs with some admixture from $n = 4$ layers, as evidenced by the photo-luminescence (PL) spectrum plotted in Fig. 1b (CW excitation at 532 nm).

To confirm strong coupling conditions in our coupled perovskite-cavity system, we performed white light (WL) reflectivity studies on our devices. In this case, a planar gold mirror (35 nm gold film thickness) was used as the top cavity reflector to optimize cavity transmission, thereby enhancing the visibility of the polariton modes. Figure 1c shows the result of a WL cavity tuning series where we systematically varied the cavity air gap and recorded angle-resolved WL spectra (see Methods section). The figure shows cross-sections through these spectra extracted at $k_\parallel = 0$. We observe clearly-developed anti-crossings for subsequent longitudinal mode orders (see Supplementary Sections S2 for a full range scan and S10 for additional strong coupling data on another flake from the same bulk crystal). At an air gap of 3.3 μm, we extract a light-matter coupling

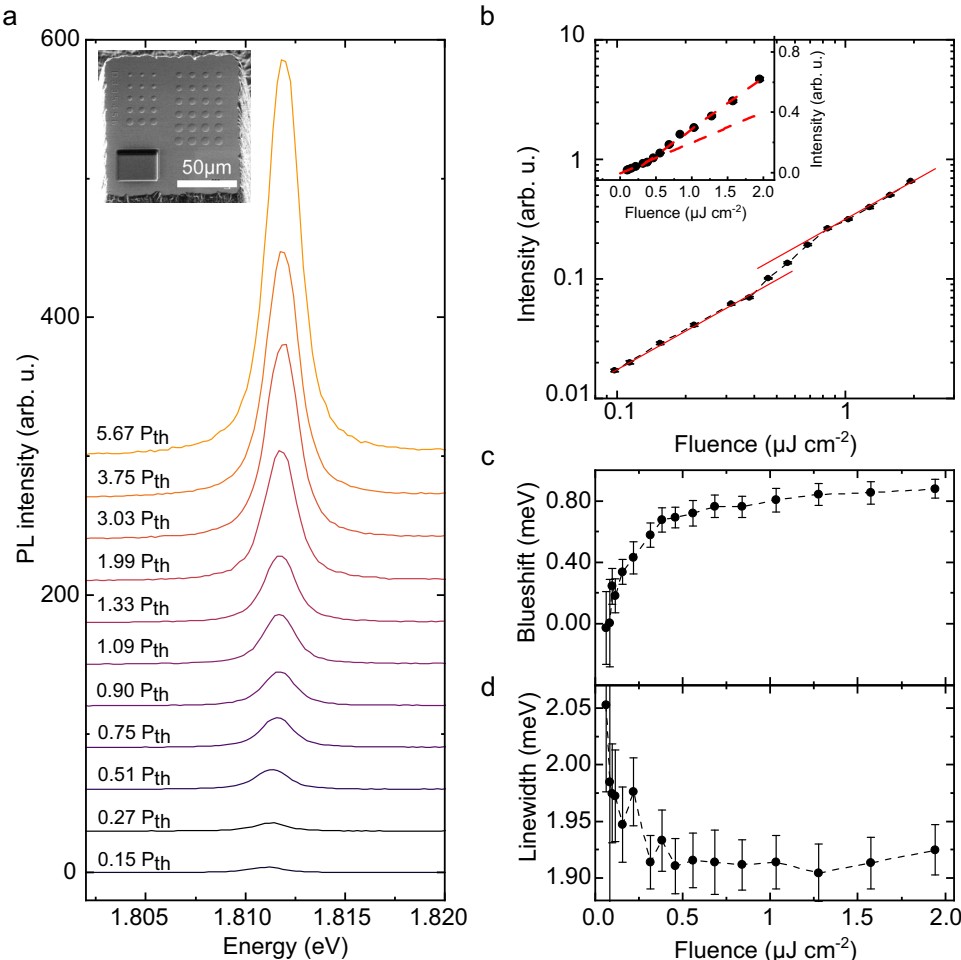

**Fig. 2 | Input-output characteristic of quasi-2D HaP polariton device. a** Power-dependent polariton emission spectra as a function of excitation pulse energy (525 nm central wavelength, 140 fs pulse duration, 80 MHz repetition rate). Spectra are vertically offset for better visibility. Excitation powers are indicated on the left in multiples of lasing threshold. **Inset**: Electron micrograph of the upper cavity distributed Bragg reflector (DBR) shaped into a $(100\,\mu m)^3$ mesa to allow short cavity distances. Sphere cap-shaped indentations ($6\,\mu m$ diameter, 150 nm depth) result in three-dimensionally confined, discrete Laguerre-Gaussian type optical modes that we couple to the halide perovskite (HaP) flake. **b** Double logarithmic plot of the input-output curve extracted from the areas under Voigt fits to the spectra in a. **Inset**: Linear representation of the same data as in the main panel. From linear fits (dashed red) we extract a polariton lasing threshold of $0.41\,\mu Jcm^{-2}$ per pulse. **c** Spectral position of the Voigt fit as a function of pulse energy relative to the first data point. The emission undergoes a blueshift continuing beyond the lasing threshold. **d** Spectral linewidth extracted from the Voigt fits as a function of pulse energy. Error bars in b-d correspond to 95% confidence intervals of the fit.

strength of $g = 23.5$ meV based on a coupled oscillator model (dashed lines in panel c). The magnitude of the coupling strength is in excellent agreement with transfer matrix simulations based on dielectric functions for the quasi-2D HaP taken from ref. 8 (see Supplementary Sections S2, S3 for details).

To reach the quantum degenerate regime of polariton condensation, we use the sphere cap-shaped indentations in the top DBR, resulting in three-dimensionally confined, Laguerre-Gaussian-type optical modes locally coupled to the perovskite sample. The confinement is instrumental in reaching polariton condensation in our system, since it enhances stimulated scattering[47] (see Supplementary Section S12 for direct comparison to a planar cavity). This strategy has been exploited before, for example, for demonstrations of polariton lasing in monolithic III-V semiconductor cavities under transverse confinement from micropillars[48]. For excitation, we used 140 fs long laser pulses at 525 nm central wavelength and 80 MHz repetition rate. In Fig. 2a, we display a power-dependent series of PL spectra obtained with the lower polariton tuned to ~1.81 eV. We notice that this energy is slightly below the energy of the weakly coupled $n = 4$ exciton (cf. Supplementary Sections S4, S10), which thus can contribute to the

polariton population via intra-cavity pumping. As we increase the power by an overall factor of 20, we observe a pronounced rise in PL power by almost three orders of magnitude. From this data, we extract the input-output characteristic shown in Fig. 2b by fitting Voigt lines (see Methods section) to the PL spectra. The non-linear increase in power displays the typical non-linear S-shape, observed in polariton lasing. A linear representation of the data is shown as an inset. From that plot, we extract a polariton lasing threshold of $P_{th} = 0.41\,\mu Jcm^{-2}$ per pulse at a spot diameter of 1.44 μm. Under otherwise equal conditions, the optical non-linearity is not observed for pure $n = 3$ phase quasi-2D HaP (see Supplementary Section S11), underpinning the importance of the weakly coupled $n = 4$ phase in our case. From the Voigt fits to the PL spectra in Fig. 2a, we also extract the central energy and emission linewidth plotted in Fig. 2c, d. We observe a blueshift of 0.84 meV extending across the threshold. Due to the femtosecond pulsed excitation, the expected collapse of the polariton linewidth is not particularly pronounced but nevertheless clearly visible[49]. We confirm the corresponding emergence of temporal coherence of the emission in the time-domain below (cf. Fig. 3). The increase in temporal coherence and associated linewidth narrowing observed for

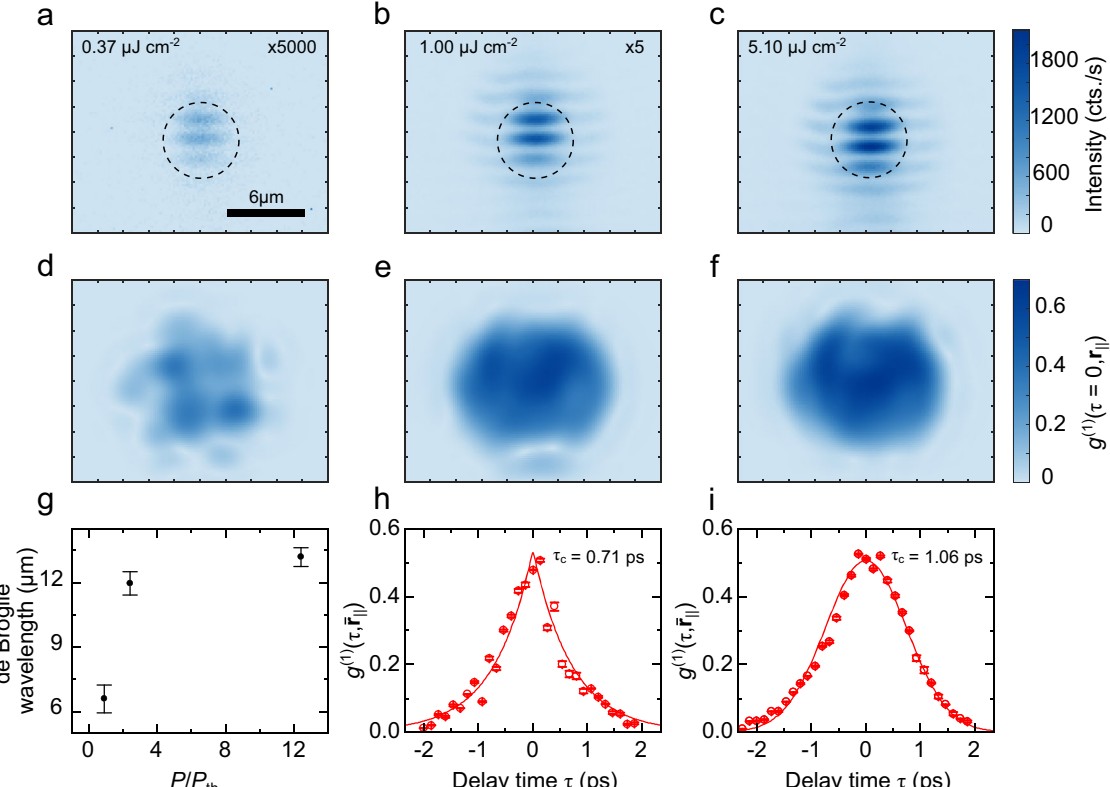

**Fig. 3 | Spatial and temporal coherence measurements.** Spatial coherence of the emission from the fundamental Laguerre-Gaussian cavity mode for different excitation pulse energies: **a** below threshold ($0.9P_{th}$), **b** above threshold ($2.4P_{th}$), **c** far above threshold ($12.4P_{th}$). Numbers in the upper right indicate multiplication factors for better visibility. **a–c** Spatially resolved interference images obtained by overlapping the emission from the mode with a spatially inverted, momentum-shifted copy realized with a Michelson interferometer using a retro-reflector in one arm. **d–f** Spatial first-order correlation function at zero delay $g^{(1)}(\tau = 0, \mathbf{r}_{||})$

extracted by Fourier analysis of **a–c**. **g** Effective de Broglie wavelength extracted from panels **d–f** by a Gaussian fit to the spatial first-order correlations within the extent of the sphere cap-shaped indentation (dashed circle in **a**). **h, i** Temporal first-order correlation function $g^{(1)}(\tau, \bar{\mathbf{r}}_{||})$ averaged over $2\,\mu m \times 2\,\mu m$ at the center of the spatial coherence map in **e, f**, as a function of interferometer delay. The fits show coherence times of 0.71 ps and 1.06 ps, respectively. Error bars in (**g**) correspond to 95% confidence intervals of the fit; in (**h, i**) they correspond to the standard deviation of the averaged $g^{(1)}$ function.

trapped polariton systems driven above threshold, as shown in our case, is usually assigned to dominating bosonic final state stimulation, as has been discussed, e.g., in refs. 48,50–52. We assign the Gaussian contribution of the Voigt lineshape to a combination of dynamic broadening (via polariton blueshift) and dynamic fluctuations of the cavity length. From the power density of the excitation at threshold, we estimate an injected density of electron-hole pairs of $\rho_{th} = 1.2 \cdot 10^{10}\,cm^{-2}$ (see Supplementary Section S5). The onset of polariton condensation thus occurs one order of magnitude below the Mott density of the $n = 3$ quasi-2D HaP[11–14]. To rule out thermal effects as the origin of the observed blueshift, we measured the temperature dependence of the quasi-2D HaP PL, observing a pronounced thermal redshift (see Supplementary Section S6).

The blueshift of the lower polariton energy stems from several non-linear processes specific to the strong light-matter coupling regime. It is induced by the exciton-exciton (X-X) scattering due to Coulomb interactions and the non-linear saturation due to Pauli exclusion[53,54]. Our aim here is to provide an indicative range of non-linear scattering coefficients by an order of magnitude comparison between reasonable theoretical estimates and the experiment. The X-X interaction constant can be estimated as $1.0\,\mu eV\mu m^2$ in a quantum well geometry, translating into the corresponding blueshift contribution for a given density and coupling (see Supplementary Section S7). The contribution of non-linear saturation can be estimated from the phase space filling, which scales as $\hbar\Omega A\sum_k |\phi(k)|^4 \approx 2.7\,\mu eV\mu m^2$,

where $\hbar\Omega = 2g$, $A$ and $\phi(k)$ are the Rabi splitting, the area of the sample and the exciton wavefunction in momentum space. Combining the two contributions, we estimate the non-linear interaction coefficient for the lower polaritons in the strongly red-detuned configuration to be $0.34\,\mu eV\mu m^2$. This agrees within approximately an order of magnitude with the experimentally observed non-linearity of $5.4\,\mu eV\mu m^2$ deduced from Fig. 2c (see Supplementary Section S7 for details). Note that for densities above $\rho_{th}$, we observe a decrease in the slope of the lower polariton energy as the density increases, which is a typical feature of the onset of a polariton condensate.

We furthermore note that the polariton lasing observed here is distinct from spatially localized defect lasing observed at low temperatures in ref. 40. We used the inherent tunability of our open cavity to verify that polariton lasing occurs with virtually identical threshold and very similar energies in multiple places of the same flake (see Supplementary Section S8 for a second input-output dataset). We also find that the measured non-linear scattering coefficient consistently remains within the same order of magnitude when comparing different places on the same flake. Observed variations in the coefficient are accounted for by its dependence on Hopfield coefficients and by slight modifications in measured effective area as well as structural composition of the flake (see Supplementary Section S7, S8 for details).

To probe the emergence of long-range order and spatio-temporal coherence of the polariton mode[39,55–57], we image the emitted PL under varying pumping conditions and quantify its spatial and temporal first-

order coherence. This is achieved by inserting a Michelson interferometer into the detection path and placing a retroreflector at the end of one interferometer arm. This results in two spatially inverted, time-delayed copies of the emission overlapped at the exit of the interferometer. Adjusting the lateral offset of the retroreflector from the optical axis introduces an offset between the copy images in momentum space. When forming the PL image on a camera, this offset causes interference fringes from which we deduce the degree of spatial first-order coherence $g^{(1)}(\tau, \mathbf{r}_{\parallel})$ as a function of position $\mathbf{r}_{\parallel}$ and interferometer delay $\tau$. Figure 3 panels a-c show the interference patterns for three powers below (panel a, $P = 0.9P_{th}$), above (panel b, $P = 2.4P_{th}$), and far above threshold (panel c, $P = 12.4P_{th}$). Note that for better visibility, the intensities in panels a and b were multiplied by factors of 5000 and 5, respectively. The dashed black circles mark the spatial extent of the sphere cap-shaped mirror indentation, i.e., they give a rough estimate of the spatial extent for the uncoupled Laguerre Gaussian photonic mode that we couple to the sample. We observe that the coherent condensate substantially spreads outside the lens. By Fourier filtering, we extract the coherent portion of the signal from these images. The first order coherence is then obtained via the relation $g^{(1)}(\tau, \mathbf{r}_{\parallel}) = \tilde{I}(\tau, \mathbf{r}_{\parallel})/2\sqrt{I_M(\tau, \mathbf{r}_{\parallel})I_R(\mathbf{r'}_{\parallel})}$ [39,58] with $\tilde{I}(\tau, \mathbf{r}_{\parallel})$ the Fourier-filtered interference pattern, $I_M(\tau, \mathbf{r}_{\parallel})$ the time-delayed image from the interferometer arm containing a planar mirror at its end and $I_R(\mathbf{r'}_{\parallel})$ the spatially inverted image from the other arm containing the retroreflector. The results of this calculation are shown in Fig. 3d–f. We see a marked increase in the degree of spatial coherence between panels d and e, when increasing the excitation power above threshold. Above threshold, $g^{(1)}$ continues to increase slowly with power up to a maximum value of ~0.7 in panel f. To further quantify the coherence properties of the condensate, we extract $g^{(1)}$ above threshold from the maps in panels e and f and observe its evolution as a function of interferometer delay $\tau$. The results are plotted in Fig. 3h, i. At $P = 2.4P_{th}$ (panel h), the first-order correlation function decays exponentially with a characteristic time scale of $\tau_c(2.4P_{th}) = 0.71$ ps.

Far above threshold at $P = 12.4P_{th}$ (panel i), the shape of the decay notably changes and is well described by a Gaussian via $g^{(1)}(\tau, \bar{\mathbf{r}}_{\parallel}) \propto \exp(\tau^2/\tau_c^2)$. This change is usually associated with number fluctuations in the condensate, causing interaction-induced energy variations[45,59]. With this model, we obtain a 50% increase in coherence time far above $P_{th}$, resulting in $\tau_c(12.4P_{th}) = 1.06$ ps. From Gaussian fits to cross-sections through the maps in Fig. 3d–f, we extract the spatial first order coherence length of the condensate within the trap. This length is defined via the standard deviation $\sigma$ of the Gaussian fit and can be interpreted in terms of an effective wavelength $\lambda_{eff} = 2\sqrt{2\pi}\sigma$, in analogy to the thermal de Broglie wavelength of a thermalized dilute Bose gas[60]. The de Broglie wavelength of our condensate is plotted in Fig. 3g, it doubles from $(6.6 \pm 0.7)$ μm below threshold to $(13.2 \pm 0.4)$ μm for pump fluences above the polariton condensation threshold.

## Discussion

Our study establishes quasi-2D lead-based halide perovskites as a particularly well-suited, versatile material for non-linear polaritonics at room temperature. We confirm polariton condensation at ambient conditions via the non-linear increase of quasi-particle population at exciton densities below the Mott density and the observed energy blueshift alongside with the onset of spatial and temporal coherence of the optical mode.

Quasi-2D halide perovskites stand out in the landscape of polariton material platforms due to an unprecedented combination of key advantages: Being chemically synthesized multi-quantum well stacks, they retain the unique flexibility in emission wavelength and oscillator strength of conventional QW stacks via controlling the thickness and number of inorganic layers. Choosing and doping the inorganic spacers even controls the exciton binding energy as a separate parameter[24]. This set of advantages is paired with excellent room-

temperature operability also found in their bulk HaP counterparts, which have recently emerged as a viable platform for monolithic room-temperature polariton devices[16–20]. However, being van-der-Waals materials, quasi-2D HaPs stand out against their bulk counterparts, since quasi-2D HaPs offer the benefit of easy hetero-integration typically found in TMDCs. This opens an exciting new field of research for room-temperature polariton condensates based on hybrid heterostructures[28], integrated with electrically tunable cavities[61] or subjected to tunable photonic lattices[42,44,62]. First demonstrations of charge-tunable devices in quasi-2D HaPs[29] clearly hint at the possibility of implementing electrically driven polariton lasers based on quasi-2D HaPs with potentially unprecedented flexibility in terms of emission wavelength and in-situ tunability. Based on the fast-paced progress with HaPs in the fields of photovoltaics and optoelectronics, we anticipate that the implementation of electrically driven room-temperature polariton devices using quasi-2D HaPs is realistic in the near future, resulting in vast opportunities for on-chip integration.

## Methods

### Perovskite synthesis

Butylammonium, $C_4H_9NH_3$ (BA) methylammonium, $CH_3NH_3$ (MA) lead iodide $(BA)_2(MA)_2Pb_3I_{10}$ ($C_4N_3$) was crystallized using the slow-cooling method with minor modifications[7,9,10] (see Supplementary Section S1 for detailed procedure and optical sample image).

### Sample preparation

DBR mirror coatings were prepared by HF sputtering of alternating layers of $SiO_2$ (107 nm) and $TiO_2$ (67 nm) terminating in $SiO_2$ with 10 mirror pairs at the bottom and 8 pairs at the top. Sphere cap-shaped indentations in the top DBR (6 μm diameter, 150 nm depth) were prepared by focused ion beam lithography (FIB, FEI Helios 600i) before mirror deposition (see Fig. 2a, inset for an SEM image).

$(BA)_2(MA)_2Pb_3I_{10}$ single crystals were micromechanically exfoliated with the scotch tape method, followed by PDMS dry stamping. We first deposit a 10 nm thick hBN bottom layer (2D Semiconductors, substrate at 100 °C, 5 min contact time), followed by the quasi-2D HaP flake (substrate at 20 °C, 20 min contact, PDMS GelPak grade zero, pre-baked at 85 °C for 4 h) and another 10 nm thick hBN flake to fully encapsulate the flake (substrate at 20 °C, 20 min contact). The preparation was performed under yellow light, and the finished samples were stored in the dark under $N_2$ atmosphere.

### Optical setup

Supplementary Fig. S5 shows a schematic of the experimental setup. Femtosecond excitation pulses are derived from a Ti:Sapphire mode-locked laser (Chameleon, Coherent) operated at 1050 nm and frequency-doubled in a BBO crystal (Crysmit optics, 21.3° cut, 0.1 mm thickness). Laser pulses are tightly focused onto the sample with a microscope objective (50X Mitutoyo Plan Apo NIR HR 0.65NA) through which we also collect the PL emission and measure WL reflectivity (Thorlabs SLS301). Details on the open cavity setup can be found in refs. 41,62. PL emission was analyzed with a Peltier-cooled CCD camera (Andor iXon Ultra 888) connected to a monochromator (Andor Shamrock 500i). Angle-resolved measurements were realized by imaging the back-focal plane of the objective onto the monochromator input with a 75X overall magnification. The Michelson interferometer consisted of a retroreflector (Thorlabs PS976M-B) in one arm and a planar mirror on a motorized stage (PI Linear Stage M-511) in the other (see Supplementary Section S9 for further details).

### Data analysis

We extract the input-output characteristic, linewidth and blueshift in Fig. 2b–d by fitting Voigt profiles to the output spectra in Fig. 2a. We approximate this profile as the weighted linear combination of a Lorentzian and a Gaussian, sharing the same center energy and linewidth.

We consistently find that an area weight of $0.69 \pm 0.03$ (standard deviation) for the Lorentzian results in the best approximation to the experimentally observed lines.

## Data availability

The experimental data underlying Figs. 1–3 have been deposited in the DARE repository under accession code https://doi.org/10.57782/JWI1KI.

## Code availability

The MATLAB code used for fitting spectra and extracting first-order autocorrelation values is available from the corresponding author upon request.

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

## Acknowledgements

The authors acknowledge support by the German Research Foundation (DFG) via the projects SCHN1376 13.1 and INST 284/234-1FUGG. Financial support by the Niedersächsisches Ministerium für Wissenschaft und Kultur ("DyNano") is gratefully acknowledged. Further support by the European Commission (ERC Project DualTwist) is acknowledged. O.K. and K.W.S. acknowledge the support from UK EPSRC Awards No. EP/X017222/1. M.E. acknowledges funding by the University of Oldenburg through a Carl von Ossietzky Young Researchers' Fellowship. H.P.A. acknowledges funding by the University of Valencia through the Margarita Salas grant (MS21-181) for the training of young doctors. D.C. and N.P.J. (present address, JNCASR, Jakkur, Bengaluru, India) thank the Ministry of Science and Culture (MWK) of the State of Lower Saxony and the Volkswagen Foundation ("Niedersächsisches Vorab - Research Cooperation Lower Saxony-Israel") and the Weizmann Institute of Science for financial support, and Dr. Sigalit Aharon (Weizmann Inst. and Princeton Univ.) for sample selection, logistics, handling, and preparation guidance for the measurements. F.E. acknowledges support through the DFG CRC 1375 NOA/B3.

## Author contributions

This work was initiated by M.E. and C.S. The experiments were conducted by M.S., C.B., and H.P.A. with support from H.S., N.M., and J.-C.D und the supervision of M.E. Experimental data were analyzed by M.S. and C.B. Quasi-2D HaP bulk crystals were grown by N.P.J. under the guidance of D.C. The top mesas were provided by F.E., and gold coatings were done by S.S. The theoretical modeling was performed by K.W.S. and O.K. The manuscript was written by M.S., C.S. and M.E. with input from all the co-authors.

## Funding

## Competing interests

The authors declare no competing interests.
