## [Peer Review file · Nature Communications]

Room-temperature polariton condensate in a quasi-2D hybrid perovskite

Corresponding Author: Dr Martin Esmann

Version 0:

Reviewer comments:

Reviewer #1

(Remarks to the Author)

In this manuscript, the authors report polariton condensation in a two-dimensional perovskite open microcavity at room temperature. The condensation is demonstrated through nonlinear power-dependent PL intensity and g_1 coherence measurements. A theoretical model is also presented to explain the observed energy blueshift in the experiment. Polariton condensation in 2D layered perovskites at low temperature has been previously reported in ref [31], and achieving condensation at room temperature with a low threshold in this work represents another major advancement. I would recommend its publication if the authors can address the following comments:

1. An absorption spectrum should be provided to accurately determine the exciton energies of the $n=3$ and $n=4$ layered halide perovskites. Additionally, further discussion is needed on the contribution of strong coupling from the $n=4$ component.
2. The condensation energy in Fig. 2a is observed at ~ 1.8 eV, yet strong coupling dispersions around this energy are not clearly shown in the manuscript. I recommend the authors measure and demonstrate the strong coupling behavior of the polaritons near the condensation energy.
3. The condensation threshold in this work is reported as 6.76 fJ per pulse at a spot diameter of 1.44 μm , corresponding to a pulse energy density of ~ 0.4 $\mu\text{J}/\text{cm}^2$. This value is much lower than the condensation threshold reported in ref. [31] at low temperature but is very close to the threshold for defect lasing described in ref. [31]. Could the authors provide more experimental evidence to justify the finding and discuss their results more in the manuscript, so that readers can be convinced that the low threshold are from polariton condensation rather than defect lasing?
4. In S4, the coupling efficiency of the pumping laser is estimated to be 20% based on the $\text{NA}=0.65$ objective. However, this assumption is valid only if the laser beam fully covers the back aperture of the objective. A direct measurement of the transmission of the pumping laser through the top DBR would provide a better estimate.
5. In Fig. S3c, the y-axis range should be optimized to provide a clearer display of the linewidth changes.
6. There are a few quite related publications on room-temperature perovskite polaritons that could be included in the introduction or outlook, such as [Nat. Nanotechnol. 19, 1283-1289 (2024), Nat Commun 13, 7388 (2022), Nat. Mater. 21, 761-766 (2022)].

Reviewer #2

(Remarks to the Author)

In their submitted paper, Esmann (or Struve) et al. claim the onset of a polariton condensate at room temperature in 2D perovskite. The measurements reported in support of their thesis are only the nonlinear increase in the total emission intensity and $g(1)$, which are not sufficient to demonstrate the polariton condensate.

In particular, it is necessary to experimentally show at least these two phenomena:

- i) The far-field emission below and above the threshold to show the collapse of the entire population;
- ii) The nonlinear shrinkage of the FWHM as a function of the pumping fluence, although the graph reported in Fig. 2d is difficult to interpret.

Moreover, a total blueshift of 0.36 meV, with an emission FWHM of ~ 1.65 meV, is too small to demonstrate a real trend. How to experimentally demonstrate lasing or a polariton condensate is well reported in this comprehensive paper (Samuel et al., 'How to Recognize Lasing,' Nat. Phot., 2009). Everyone should follow these guidelines. Furthermore, in this very recent paper (<https://doi.org/10.1038/s41563-024-01980-3>), the authors successfully exploit the polariton condensate at room temperature, although utilizing 3D perovskite.

For these reasons, I cannot recommend the publication of the paper at this stage in Nature Communications. The authors should provide either an unambiguous demonstration of the polariton condensate or unique properties of their effects for reevaluating the manuscript in a high-impact factor journal like Nature Communications.

Other comments:

- 1) There is confusion regarding the names and order of the authors. Please check.
- 2) The figures are not cited in order in the main text.
- 3) Some citations are wrong, the perovskite used in [31] is not in (BA)₂PbI₄;
- 4) For comparison with the paper reported in literature on this framework until now (<https://doi.org/10.1038/s41567-019-0764-5> / <https://doi.org/10.1021/acs.nanolett.1c01162> / <https://doi.org/10.1515/nanoph-2023-0829>), the threshold should be reported as fluence (Joule/cm²) or power (W). The advantage of having a polariton condensate is its very low threshold, which is not clear in this work because it is expressed in femtojoules (fJ) and cannot be directly compared with the values reported in the literature.
- 5) Fig. 1b reveals the presence of two phases in the crystal. Efforts could be made to improve the quality of the crystal. Additionally, real-space images of the crystals are not reported.
- 6) Fig. 2d does not show a reduction in the FWHM. This graph does not make sense.
- 7) Why is the fringe pattern visible even below the threshold? How homogeneous is the sample?

Reviewer #3

(Remarks to the Author)

In the manuscript titled "Room-temperature polariton condensate in a two-dimensional hybrid perovskite," Struve et al. report the observation of polariton lasing at room temperature from layered perovskite materials embedded in a vertical microcavity. By monitoring the reflectivity spectra while scanning the exciton-photon detuning, the authors demonstrate strong coupling with evidence of the avoided-crossing effect between perovskite excitons and the photonic mode. Lasing emission was observed when the planar cavity was replaced by a 0D cavity confinement, induced by sphere-cap-shaped structures in the top mirror. The authors analyzed the nonlinearity (via energy blueshift) and spatial coherence (via interferometry measurement) of the lasing emission. Based on their measurements, the effects are attributed to polariton condensation.

While the results are scientifically sound, I have several concerns with this work that prevent me from recommending it for publication:

1. The reported lasing effect is photon lasing, as the excitonic fraction from the polaritonic mode is extremely small. Indeed, the excitonic fraction is about 1.3% if we use the light-matter coupling strength of $g = 23.5$ meV, an exciton energy of 2 eV, and a photon energy of 1.8 eV. Thus, the lasing emission is 98.7% photonic-like if attributed to polariton lasing. We should not expect any interesting polaritonic features from such a low excitonic fraction.
2. The nonlinearity investigation is based on an energy blueshift ranging from 0 to 0.36 meV (Fig. 2c), while the polariton linewidth is about 1.6 meV, several times larger. Therefore, the nonlinearity presented here is not significant. Moreover, the blueshift in perovskite material emission could also be attributed to thermal effects.
3. Regarding the coherence measurement, the same effect is expected from pure photon lasing. Since the lasing emission originates from a well-defined photonic 0D mode, spatial coherence is naturally expected within the spatial profile of this mode (which could extend beyond the shape of the sphere cap due to shallow confinement).

Additionally, I have concerns regarding the novelty of this work, which I do not find suitable for Nature Communications. Room-temperature polariton lasing has already been reported in perovskites by other groups, with even stronger nonlinear effects such as optical parametric oscillation and superfluidity. In terms of materials, previous reports used 3D inorganic perovskites and 3D hybrid perovskites, and I do not see how using layered perovskites introduces novelty. Other groups have reported polariton lasing with Rabi splittings in the range of hundreds of meV, an order of magnitude higher than what is reported here. This indicates that the perovskite material used in this work has much weaker oscillator strength.

Additional comments:

4. The authors refer to their perovskite as "2D perovskite," which is incorrect. It should be termed "layered perovskite" since $n = 3$, not the 2D perovskite case where $n = 1$.
5. I am curious why the Rabi splitting is so small in this work, as typical Rabi splitting in perovskite materials (2D, 3D, hybrid, all-inorganic) is around 100-200 meV.

Version 1:

Reviewer comments:

Reviewer #1

(Remarks to the Author)

I believe the authors have adequately addressed my previous comments in the revised manuscript. Based on the

improvements made, I am pleased to recommend its publication in Nature Communications.

Reviewer #2

(Remarks to the Author)

I appreciate the authors for addressing my concerns, providing additional experimental results, and offering further interpretations of the data. The revised version of the manuscript is notably improved and provides a clearer explanation of the measured effect.

I still have a couple of concerns.

- 1) Can the authors experimentally demonstrate the role of the $n=4$ phase in polariton lasing? It would also be useful to include a comparison with a sample where the $n=4$ phase is absent.
- 2) What is the physical reason for the weak coupling between the $n=4$ phase and the optical mode?
- 3) The actual effect of the 0D confinement induced by the sphere on the DBR is not entirely clear. Does this effect also occur in the planar section? How does it impact the polariton threshold—does it change or disappear?

Reviewer #3

(Remarks to the Author)

The authors have addressed most of my questions from the previous report and have provided substantially improved data along with additional experiments. I now believe that the reported results do correspond to polariton condensation. However, I would recommend publication of this work in Nature Communications only after another revision, as I still have several concerns regarding the new data set:

1. The authors present new measurements with an excitonic fraction of 1.5%, compared to 1.3% previously—an increase of about 1.15 times. How can such a relatively small change result in a twofold increase in the blueshift?
2. Figure 2d, which shows the linewidth narrowing, is not discussed in the manuscript. In the case of polariton condensation in a planar cavity, linewidth narrowing above threshold is attributed to macroscopic occupation at $k = 0$, with negligible population at other momenta in the dispersive cavity. In contrast, for photon lasing in a trapped (i.e., nondispersive, discrete) state, narrowing is due to the dominance of stimulated emission in the lasing mode over spontaneous emission into other channels. However, the physical origin of linewidth narrowing in the case of polariton condensation in a trapped state remains unclear to me. Can the authors clarify this point?
3. With the improved linewidth extraction, can the authors now quantify the contributions to the linewidth: how much arises from the photonic mode, how much from the excitonic component, and what is the contribution from inhomogeneous broadening of the exciton?

Version 2:

Reviewer comments:

Reviewer #2

(Remarks to the Author)

As indicated in my previous remarks, I am pleased to recommend the paper for publication in Nature Communications.

Reviewer #3

(Remarks to the Author)

In the revised manuscript and response letter, the authors addressed all my concerns. I therefore recommend the publication of the manuscript in Nature Communications.

Blue italics: Reviewer comment

Black: Our response

Red: Changes to the manuscript/SI

Reviewer #1 (Remarks to the Author):

In this manuscript, the authors report polariton condensation in a two-dimensional perovskite open microcavity at room temperature. The condensation is demonstrated through nonlinear power-dependent PL intensity and g_1 coherence measurements. A theoretical model is also presented to explain the observed energy blueshift in the experiment. Polariton condensation in 2D layered perovskites at low temperature has been previously reported in ref [31], and achieving condensation at room temperature with a low threshold in this work represents another major advancement. I would recommend its publication if the authors can address the following comments:

We thank the reviewer for the careful and very positive assessment of our work. We are particularly pleased with the reviewer's opinion that the demonstrated room-temperature polariton condensation represents a major advancement.

Below, we respond point-to-point to each of the questions raised by the reviewer. We in particular include **new experimental datasets (in particular Fig. 2 of the main text)** to better illustrate the main claims of the manuscript.

1. An absorption spectrum should be provided to accurately determine the exciton energies of the $n=3$ and $n=4$ layered halide perovskites. Additionally, further discussion is needed on the contribution of strong coupling from the $n=4$ component.

To determine the exciton energies more precisely, **we measured additional PL and reflectivity spectra shown below.**

We now show Fig. R1 and the corresponding description in the new Supplementary Section S10 as a new Fig. S9.

Figure R1: Photoluminescence (PL, blue) and white light (WL, red) reflectivity spectrum of a mixed phase layered perovskite flake derived from the same bulk crystal used for the polariton condensation experiments. Two resonances corresponding to the $n=3$ and the $n=4$ exciton are clearly visible in both PL and WL reflectivity.

We exfoliated a new HaP flake from the same bulk crystal that was previously used for the samples in our polariton condensation measurements. The crystal has contributions from both $n=4$ and $n=3$ phases. In Fig. R1, we show a photoluminescence (PL, blue) and a white light (WL, red) reflectivity spectrum of the flake after it was transferred onto a DBR with identical properties compared to the DBRs used for the measurements in the main text. The PL shows two distinct peaks at 1.995 eV and 1.92 eV respectively, corresponding to the $n=3$ and $n=4$ phase exciton of the crystal. The WL reflectivity spectrum shows a dip at 1.99 eV, corresponding to the $n=3$ exciton and a significantly shallower absorption dip at ~ 1.93 eV, corresponding to the $n=4$ exciton. WL and PL are in close agreement for $n=3$, while the discrepancy of 10 meV for $n=4$ results from the uncertainty in determining the position of the shallow WL reflectivity. However, both pairs of PL peak/ WL dip are clearly matching within their linewidth.

We further address the contribution of strong coupling from the $n=4$ component as part of our answer to point 2 below.

2. The condensation energy in Fig. 2a is observed at ~ 1.8 eV, yet strong coupling dispersions around this energy are not clearly shown in the manuscript. I recommend the authors measure and demonstrate the strong coupling behavior of the polaritons near the condensation energy.

We measured additional strong coupling data of a freshly prepared flake within the 1.78 to 2.05 eV range, covering the polariton condensation energy.

- **We extended the energy span of the full range strong coupling dispersion in Supplementary Fig. S2 (shown as Fig. R2 below).**
- **We now show additional strong coupling data in Fig. R3 below and include it in the new Supplementary Section S10 as Fig. S10 together with the corresponding description.**
- **We describe and reference the new supplementary section S10 in the main text.**

The strong coupling leads to an avoided crossing centered at the energy of the exciton, which is in our case the $n=3$ exciton at 2 eV. The spectral range accessible in our strong coupling measurements was constrained by the spectral measurement window of our spectrometer. For the demonstration of strong coupling, we selected a range that centers the exciton in the spectrum as shown in Fig. S2.

We address the reviewer’s question in two parts:

- First, we extended the corresponding transfer matrix simulation in Fig. S2 down to energies of 1.75 eV. The revised version of Fig. S2 is included here as Fig. R2.
- Second, we also investigated the strong coupling behavior of a freshly prepared flake within the 1.78 to 2.05 eV range. The data is shown as Fig. R3 below and now included as Fig S10 in the revised Supplementary information. At approximately 2 eV, a Rabi splitting associated with the $n=3$ phase HaP exciton is observed as before, with a coupling strength of 16 meV. The reduced coupling strength compared to the data in Fig. 1 of the main text is attributed to the reduced flake thickness, which we estimate to be around 200 nm. Importantly, no evidence of a Rabi gap is observed at 1.92eV, the energy of the $n=4$ phase HaP exciton. Instead, the data reveals a clear dispersive behavior attributed to the variation in the real part of the refractive index. This is in agreement with the much lower absorption for $n=4$ shown in Fig. R1. We note that the dispersive feature is almost absent in the device discussed in the main text, which hints at an even lower contribution from the $n=4$ phase. These results lead to the conclusion that the $n=4$ HaP phase does not couple strongly to the cavity photon and can hence act as an effective intra-cavity pump as discussed in the main text and supported by the absence of strong coupling features around 1.92 eV in the PL dispersion shown in Fig. S3.

Figure R2: Revised Fig. S2 of the supplementary information. The energy range of the transfer matrix simulation in panel b has been extended.

Figure R3: Strong coupling measurement of a layered perovskite flake with a thickness of 200 nm. At 2 eV, the $n=3$ exciton shows strong coupling with the cavity photon, the $n=4$ exciton at 1.92eV only results in a dispersive feature and no Rabi gap. The dispersive feature is almost absent in the device discussed in the main text, for which full strong coupling data are shown in Fig. S2. This hints at an even lower $n=4$ contribution for the device used for polariton condensation experiments and is in full agreement with the absence of a Rabi gap at 1.92eV in the PL data presented in Fig. S3 of the Supplementary Information.

The reference in the main text to the new data in Fig. R3 reads:

“We observe clearly-developed anti-crossings for subsequent longitudinal mode orders (see Supplementary Sections S2 for a full range scan and S10 for additional strong coupling data on another flake from the same bulk crystal).”

3. The condensation threshold in this work is reported as 6.76 fJ per pulse at a spot diameter of 1.44 μm , corresponding to a pulse energy density of $\sim 0.4 \text{ fJ}/\text{cm}^2$. This value is much lower than the condensation threshold reported in ref. [31] at low temperature but is very close to the threshold for lasing described in ref. [31]. Could the authors provide more experimental evidence to justify the finding and discuss their results more in the manuscript, so that readers can be convinced that the low threshold are from polariton condensation rather than defect lasing?

We thank the reviewer for the pertinent comment. Indeed, the observed threshold in our work is fairly low, which we think is a consequence of the effective intracavity pumping provided by the $n=4$ portion of the layered perovskite.

- **To make a clear distinction from defect lasing, we now included the key point of the discussion below in the main text of the manuscript.**

There are in fact two arguments that speak against defect lasing in our experiment:

- On the one hand, Ref. [31] (now Ref [39] in the main text) conducts experiments at cryogenic temperatures of 4 K. At those temperatures, defects are quite accessible; they provide gain and are easy to saturate, which facilitates lasing. Our experiments are conducted at room temperature. While in Ref. [31] (now [39]) the defect binding energies are not explicitly mentioned, it is unlikely that they exceed 25meV and, hence, we conclude that at room temperature defects in our material do not efficiently trap carriers that provide sufficient gain to support a lasing transition.

- More importantly, the defects in Ref. [31] (now [39]) were spatially localized. Our open cavity system gives us the opportunity to systematically measure polariton condensation at different spots of a sample. **We used this to verify that polariton lasing occurs with virtually identical threshold and very similar energies on multiple places of the same flake. In fact, we used one of these new datasets to replace the data in Fig. 2 of the main text**, since the reduction in linewidth across the polariton threshold was much better visible than before and the exciton Hopfield coefficient was larger, resulting in a larger blue shift. This spatially homogenous lasing behavior of our material allows us to conclude that localized defects do not dominate the observed polariton lasing.

We attribute the low threshold to the contribution of the $n=4$ phase. By tuning the polariton energy through the open cavity system below the excitonic absorption, the weakly coupled $n=4$ phase works as an intra cavity pump that further populates the polariton states, lowering the threshold significantly.

The new paragraph in the results section of the main text now reads:

“We furthermore note that the polariton lasing observed here is distinct from spatially localized defect lasing observed at low temperatures in Ref. [39]. We used the inherent tunability of our open cavity to verify that polariton lasing occurs with virtually identical threshold and very similar energies on multiple places of the same flake (see Supplementary section S8 for a second input-output dataset).”

Additionally, the bulk crystals from which we derived our samples were thoroughly pre-characterized by methods that served specifically to look for defects [PNAS 121, e2316867121 (2024)] (Ref. [10] in the revised main text). The bulk material we used was from the same batch of high quality crystals as investigated in that study.

4. In S4, the coupling efficiency of the pumping laser is estimated to be 20% based on the $NA=0.65$ objective. However, this assumption is valid only if the laser beam fully covers the back aperture of the objective. A direct measurement of the transmission of the pumping laser through the top DBR would provide a better estimate.

We extended Supplementary Section S5 based on the explanation below:

Our excitation laser is indeed matched in diameter to the full back aperture of the microscope objective. Since our cavity is based on DBRs, the transmission of excitation light through the top DBR alone will, however, be different from the coupling efficiency of the full cavity, in particular at energies that are outside the DBR stop band and sensitively depend on the Bragg fringes of the DBRs.

We provide further details on how the estimated 20% coupling efficiency was computed: Fig. R4 shows a momentum-dependent transfer matrix simulation of the reflectivity from the full cavity without active material. We chose our excitation pulse to be exactly centered in energy on the first Bragg minimum at 2.368eV for $k_{||} = 0$. Note that the spectral position of the Bragg minimum is largely independent of the cavity air gap. To estimate the coupling efficiency, we selected the in-plane momentum range for which the first Bragg minimum still falls within the FWHM of the excitation pulse. This area is marked by the white dashed rectangle in Fig. R4 and extends to

$\pm 3.6 \mu\text{m}^{-1}$. This range corresponds to an effective NA of 0.3. Comparing this value to the NA of the microscope objective of NA=0.65, we obtain a coupling efficiency of 20%.

Figure R4: Momentum-dependent transfer matrix simulation of the full cavity reflectivity without active material. The height of the white dashed area shows the FWHM of the excitation pulse, while its width marks all momenta for which excitation light can efficiently enter the cavity. This calculation results in a coupling efficiency of 20%.

We extended Supplementary Section S5 by the following paragraph:

“Note that we have chosen our excitation wavelength such that it coincides exactly with the first Bragg minimum of the cavity DBRs at $k_{||} = 0$. The spectral position of the Bragg minimum is largely independent of the length of the cavity air gap. Comparing the effective acceptance NA of 0.3 of the cavity to the used microscope objective with NA=0.65, this we obtain an estimated results in a coupling efficiency of 20%. Note that our excitation laser is matched in diameter to the full back aperture of the microscope objective.”

5. In Fig. S3c, the y-axis range should be optimized to provide a clearer display of the linewidth changes.

We followed the reviewer’s suggestion and changed the y-axis range in Fig. S3 accordingly. The revised version of Fig. S3 (now Fig. S7) is included below as Fig. R5.

Considering that we re-measured the central dataset of Fig. 2 with a much better visible non-linear linewidth behavior and higher blue shifts, we decided to move the previous version of Fig. 2 of the main text to the Supplementary Information. **Together, the previous version of Fig. 2 (now Fig. S6) and the revised version of Fig. S3 (now Fig. S7) constitute a full second dataset of input-output data, which further validate our observations.**

The fully re-measured dataset is now included in the main text as Fig. 2. We show it below as Fig. R6.

Figure R5: Revised version of Fig. S3 (now Fig. S7) with optimized y-axis range for panel c.

Figure R6: New Figure 2 of the main text with fully re-measured input-output characteristic on a different area of the same quasi-2D HaP flake. The blueshift across the polariton condensation threshold is twice as large as before and the reduction of polariton linewidth at the threshold is clearly visible. We attribute both to a larger exciton fraction of the polariton leading to enhanced interaction-driven non-linearities.

Please note that we also enhanced the accuracy of our fits by using Voigt profiles. This is described in a new part of the methods section, which reads:

“We extract the input-output characteristic, linewidth and blueshift in Fig. 2b-d by fitting Voigt profiles to the output spectra in Fig. 2a. We approximate this profile as the weighted linear combination of a Lorentzian and a Gaussian, sharing the same center energy and linewidth. We consistently find that an area weight of 0.69 ± 0.03 for the Lorentzian results in the best approximation to the experimentally observed lines.”

6. *There are a few quite related publications on room-temperature perovskite polaritons that could be included in the introduction or outlook, such as [Nat. Nanotechnol. 19, 1283-1289 (2024), Nat Commun 13, 7388 (2022), Nat. Mater. 21, 761-766 (2022)].*

We thank the reviewer for the remark. We have included these references and further related ones in the introduction and the outlook of our manuscript as new References [19,16,17].

The corresponding section in the introduction now reads:

“Unlike for their 3D counterparts, which have recently emerged as a promising material platform for monolithic room-temperature polariton devices [16-20], choosing the thickness of the inorganic layers in terms of the number of unit cells, n , results in a unique control over the exciton resonances in quasi-2D HaPs”

The corresponding section in the outlook now reads:

“This set of advantages is paired with excellent room-temperature operability also found in their bulk HaP counterparts, which have recently emerged as a viable platform for monolithic room-temperature polariton devices [16-20].”

We thank the reviewer again for the important remarks and helpful suggestions. We hope that our revised manuscript can now be accepted for publication in Nature Communications.

Reviewer #2 (Remarks to the Author):

In their submitted paper, Esmann (or Struve) et al. claim the onset of a polariton condensate at room temperature in 2D perovskite. The measurements reported in support of their thesis are only the nonlinear increase in the total emission intensity and $g(1)$, which are not sufficient to demonstrate the polariton condensate.

We thank the reviewer for assessing our manuscript. To address the important comments made by the reviewer, **we completely re-measured the central datasets of the work showing polariton condensation**. We show that the non-linear reduction in linewidth and the increase of coherence length inside the trap are now much more evident. Below, we respond point-to-point to each of the questions and demands made by the reviewer:

In particular, it is necessary to experimentally show at least these two phenomena:

i) The far-field emission below and above the threshold to show the collapse of the entire population;

Since our measurements are performed for polaritons confined in a three-dimensional trap, a distinct measurement of the collapse in k-space as seen for polaritons in a 2D cavity is problematic. Even below threshold, we find localized trap modes that have finite linewidth and dictate localization in k-space through the trap size.

As an alternative, we measured the spatial coherence of the polaritons as a function of pump fluence to show the expansion of the condensate in real space. From this, we extract the spatial coherence length inside the trap as the standard deviation σ of a Gaussian fit to the spatial first order coherence $g^{(1)}(\tau = 0, \mathbf{r}_{\parallel})$ within the trap. Based on the extracted coherence length, we are able to study the effective wavelength $\lambda_{eff} = 2\sqrt{2\pi}\sigma$ of our condensate in analogy to the thermal de Broglie wavelength of a thermalized dilute Bose gas [Roumpou et al. PNAS **109**, 6467-6472 (2012)]. For the chosen pump powers, the corresponding extracted de Broglie wavelengths are plotted in Fig. R7g below. The de Broglie wavelength of our condensate doubles from $(6.6 \pm 0.7) \mu\text{m}$ to $(13.2 \pm 0.4) \mu\text{m}$ for pump fluences above the polariton condensation threshold.

- **Fig. R7g is now included in the main text as Fig. 3g together with the corresponding discussion.**

The added section in the main text reads:

“From Gaussian fits to cross-sections through the maps in Fig. 3 d-f, we extract the spatial first order coherence length of the condensate within the trap. This length is defined via the standard deviation σ of the Gaussian fit and can be interpreted in terms of an effective wavelength $\lambda_{eff} = 2\sqrt{2\pi}\sigma$, in analogy to the thermal de Broglie wavelength of a thermalized dilute Bose gas [54]. The de Broglie wavelength of our condensate is plotted in Fig. 3g, it doubles from $(6.6 \pm 0.7) \mu\text{m}$ below threshold to $(13.2 \pm 0.4) \mu\text{m}$ for pump fluences above the polariton condensation threshold.”

Figure R7: Figure 3 of the main text including new panel g. Spatial coherence measurements of the emission from the fundamental Laguerre-Gaussian cavity mode for different excitation pulse energies: **a** below threshold ($0.9 P_{th}$), **b** above threshold ($2.4 P_{th}$), **c** far above threshold ($12.4 P_{th}$). Numbers in the upper right indicate multiplication factors for better visibility. **a-c** show spatially resolved interference images obtained by overlapping the emission from the mode with a spatially inverted, momentum-shifted copy realized with a Michelson interferometer using a retro-reflector in one arm. **d-f** show the spatial first-order correlation function at zero delay $g^{(1)}(\tau = 0, \mathbf{r}_{||})$ extracted by Fourier analysis of panels a-c. **g** shows the effective de Broglie wavelength extracted from panels d-f by a Gaussian fit to the spatial first order correlations within the extent of the sphere cap shaped indentation (dashed circle in a). **h(i)** shows the temporal first-order correlation function $g^{(1)}(\tau, \mathbf{r}_{||})$ averaged over $2\mu\text{m} \times 2\mu\text{m}$ at the center of the spatial coherence map in e(f) as a function of interferometer delay. The fits show coherence times of 0.71 ps (1.06 ps).

ii) The nonlinear shrinkage of the FWHM as a function of the pumping fluence, although the graph reported in Fig. 2d is difficult to interpret. How to experimentally demonstrate lasing or a polariton condensate is well reported in this comprehensive paper (Samuel et al., 'How to Recognize Lasing,' Nat. Phot., 2009). Everyone should follow these guidelines.

- To address this request, **we fully re-measured the input-output characteristic of Figure 2 on a different position of the same perovskite flake. This dataset shown in Fig. R8 below replaces the data previously shown in Fig. 2.**
- **The old data in Fig. 2 have instead been moved to the Supplementary as Fig. S6 to underpin the spatially homogeneous polariton lasing behavior of our sample.**

In the new data, a threshold behavior is observed at $0.41 \mu\text{J cm}^{-2}$ (6.8 fJ in the old representation, i.e. very close to the previous value). Most importantly, **we trust that the reviewer agrees with us that the drop in linewidth at threshold is now completely obvious from Fig. R8d.**

Since the excitonic fraction of the polariton was higher in these new measurements, also the blueshift observed in Fig. R8c is larger than in the data set from our previous submission.

We emphasize again, that the measured linewidths appear broadened due to the pulsed femtosecond excitation. As an alternative characterization, we instead showed the collapse of the linewidth by measuring the changes in temporal coherence profiles as shown in Fig.3 of the main text. With the new datasets in place, we now see both the spectral and temporal signatures of the linewidth collapse.

Figure R8: New Figure 2 of the main text with fully re-measured input-output characteristic on a different area of the same quasi-2D HaP flake. The blueshift across the polariton condensation threshold is twice as large as before and the reduction of polariton linewidth at the threshold is clearly visible. We attribute both to a larger exciton fraction of the polariton leading to enhanced interaction-driven non-linearities.

In addition, we demonstrate other distinct features of polariton condensation:

1. the non-linear threshold below the Mott density,
2. the increase of 1st order temporal coherence through the collapse of the linewidth,
3. the distinct increase in de Broglie wavelength of the condensate in Fig. 3g of the main text,
4. and the persistent non-linear blueshift of the condensate.

While the observed blueshift of our polariton system is smaller than the emission linewidth, the shift is statistically significant with clearly non-overlapping error bars in Fig. R8c. At the same time, our theoretical calculations indicate that the expected blueshift is inherently small. Interestingly, the experimentally measured blueshift exceeds our model, most likely due to the presence of a dark exciton population leading to additional phase-space filling effects which we did not account for in our description.

Please note that we also enhanced the accuracy of our fits by using Voigt profiles. This is described in a new part of the methods section, which reads:

“We extract the input-output characteristic, linewidth and blueshift in Fig. 2b-d by fitting Voigt profiles to the output spectra in Fig. 2a. We approximate this profile as the weighted linear combination of a Lorentzian and a Gaussian, sharing the same center energy and linewidth. We consistently find that an area weight of 0.69 ± 0.03 for the Lorentzian results in the best approximation to the experimentally observed lines.”

Furthermore, in this very recent paper (<https://doi.org/10.1038/s41563-024-01980-3>), the authors successfully exploit the polariton condensate at room temperature, although utilizing 3D perovskite.

We thank the reviewer for the suggestion.

We now cite the suggested paper (<https://doi.org/10.1038/s41563-024-01980-3>) as Ref. [20] in the main text.

There are further studies on 3D perovskites demonstrating polariton condensation. Nevertheless, we are convinced that quasi-2D HaPs possess distinct advantages over their 3D counterparts, which constitutes the novelty and impact of our work.

To elaborate this point further, we extended and clarified the discussion on novelty both in the introduction and conclusion of our revised manuscript based on the arguments below:

1. Unlike for their 3D counterparts, choosing the thickness of the inorganic layers in terms of the number of unit cells, n , results in a unique control over the exciton resonances in quasi-2D HaPs, resulting in flexible spectral emission and absorption characteristics for optoelectronic devices.
2. In quasi-2D HaPs, the exciton binding energy is a separate, directly accessible tuning parameter, e.g. via doping the organic spacer. This has no analog in 3D HaPs.
3. 2D HaPs can be micromechanically cleaved and re-assembled in a well-defined manner. This has already lead to first demonstrations of charge-tunable gated devices in 2D HaPs with electrically controlled optical response and clearly hints at the possibility of realizing electrically driven room-temperature polariton lasers based on quasi-2D HaPs with potentially unprecedented flexibility in terms of emission wavelength and in-situ tenability.
4. Being van-der-Waals materials, quasi-2D HaPs stand out against their bulk counterparts, since quasi-2D HaPs offer the benefit of easy hetero-integration typically found in TMDCs. This opens an exciting new field of research for room-temperature polariton condensates based on hybrid heterostructures, integrated with electrically tunable cavities or subjected to tunable photonic lattices.

The corresponding revised section in the introduction reads:

“[...] Unlike for their 3D counterparts, which have recently emerged as a promising material platform for monolithic room-temperature polariton devices [16-20], choosing the thickness of the inorganic layers in terms of the number of unit cells, n , results in a unique control over the exciton resonances in quasi-2D HaPs [7,21-23]. Even the exciton binding energy is a separate, directly accessible tuning parameter in quasi-2D HaPs, e.g. via doping the organic spacer [24]. Most importantly, 2D HaPs can be micromechanically cleaved and re-assembled in a well-defined manner [4,25-28]. This has already lead to first demonstrations of charge-tunable gated devices in 2D HaPs [29] with electrically controlled optical response.

The giant excitonic oscillator strengths and binding energies, alongside with spectral tunability and versatility for complex hetero-integration render quasi-2D HaPs a particularly promising material class for exploring room temperature polaritonics. The formation of cavity exciton-polaritons based on these materials has recently been observed in planar dielectric, metallic and hybrid cavities [30-36]. However, in quasi-2D HaPs, the bosonic condensation of exciton-polaritons [37,38] - their most significant feature for applications based on their quantum coherence - has only been demonstrated at 4 Kelvin in the 2D HaP (PEA)₂PbI₄ (n = 1, PEA=phenethylammonium) [39], and so far remained elusive at room temperature. [...]"

The corresponding section in the outlook reads:

"Quasi-2D halide perovskites stand out in the landscape of polariton material platforms due to an unprecedented combination of key advantages: Being chemically synthesized multi-quantum well stacks, they retain the unique flexibility in emission wavelength and oscillator strength of conventional QW stacks via controlling the thickness and number of inorganic layers. Choosing and doping the inorganic spacers even controls the exciton binding energy as a separate parameter [24]. This set of advantages is paired with excellent room-temperature operability also found in their bulk HaP counterparts, which have recently emerged as a viable platform for monolithic room-temperature polariton devices [16-20]. However, being van-der-Waals materials, quasi-2D HaPs stand out against their bulk counterparts, since quasi-2D HaPs offer the benefit of easy hetero-integration typically found in TMDCs. This opens an exciting new field of research for room-temperature polariton condensates based on hybrid heterostructures [28], integrated with electrically tunable cavities [55] or subjected to tunable photonic lattices [41,43,56]. First demonstrations of charge tunable devices in quasi-2D HaPs [29] clearly hint at the possibility of implementing electrically driven polariton lasers based on quasi-2D HaPs with potentially unprecedented flexibility in terms of emission wavelength and in-situ tunability. Based on the fast-paced progress with HaPs in the fields of photovoltaics and optoelectronics, we anticipate that the implementation of electrically driven room-temperature polariton devices using quasi-2D HaPs is realistic in the near future resulting in vast opportunities for on-chip integration."

For these reasons, I cannot recommend the publication of the paper at this stage in Nature Communications. The authors should provide either an unambiguous demonstration of the polariton condensate or unique properties of their effects for reevaluating the manuscript in a high-impact factor journal like Nature Communications.

We trust that, based on our revisions and additions, the reviewer will agree with us that the demonstration of the polariton condensation is unambiguous in our experiments.

Other comments:

1) There is confusion regarding the names and order of the authors. Please check.

The distinction between corresponding author and first author seems indeed unclear in the submission portal. We thank the reviewer for pointing this out and we will make sure to clarify the order of authors with the editorial office upon resubmission.

2) *The figures are not cited in order in the main text.*

We thank the reviewer for this remark. The figures are now cited in numerical order according to where they are mentioned first in our manuscript.

3) *Some citations are wrong, the perovskite used in [31] is not in (BA)₂PbI₄;*

We apologize for this error and now cite Ref. [31] (now Ref. [39]) correctly.

4) *For comparison with the paper reported in literature on this framework until now (<https://doi.org/10.1038/s41567-019-0764-5> / <https://doi.org/10.1021/acs.nanolett.1c01162> / <https://doi.org/10.1515/nanoph-2023-0829>), the threshold should be reported as fluence (Joule/cm²) or power (W). The advantage of having a polariton condensate is its very low threshold, which is not clear in this work because it is expressed in femtojoules (fJ) and cannot be directly compared with the values reported in the literature.*

We followed the reviewer's suggestion and now report excitation powers in terms of fluences throughout the manuscript.

5) *Fig. 1b reveals the presence of two phases in the crystal. Efforts could be made to improve the quality of the crystal. Additionally, real-space images of the crystals are not reported.*

The reviewer is correct, our crystal indeed presents two phases. However, we emphasize that the spatial distribution of the n=4 phase is homogeneous and its presence is an essential feature in our work: Surprisingly, we found that the n=4 contributions in our crystals are instrumental to the observation of polariton condensation at low threshold fluence. While the n=4 component does not couple strongly to cavity-confined photons, it however acts as an effective intra-cavity pump with the n=4 exciton positioned at 1.92 eV i.e. slightly above the energies for which polariton condensation with low threshold was observed.

To consolidate these observations further:

- **We performed a more detailed optical characterization of our quasi-2D HaP flake (Fig. R9 and R10 below) illustrating distinctly different light-matter coupling behaviors of the two crystal phases. This extended characterization is now discussed in the new Supplementary Section S10 (new Figs. S9 and S10) and referenced in the main text.**
 - **We show a real space image of our quasi-2D HaP flake (Fig. R11 below), following the request made by the reviewer. This image is now included as new Fig. S1 in the Supplementary Information.**
1. We exfoliated a new quasi-2D HaP flake from the same bulk crystal that was used for the samples in our polariton condensation measurements. The crystal has contributions from both n=4 and n=3 phases. In Fig. R9 we show a photoluminescence (PL, blue) and a white light (WL, red) reflectivity spectrum of the flake after it was transferred onto a DBR with identical properties compared to the DBRs used for the measurements in the main text. The PL shows two distinct peaks at 1.995 eV and 1.92 eV respectively, corresponding to the n=3 and n=4 phase exciton of the crystal. The WL reflectivity spectrum shows a dip at 1.99 eV, corresponding

to the $n=3$ exciton and a significantly shallower absorption dip at ~ 1.93 eV, corresponding to the $n=4$ exciton. WL and PL are in close agreement, both pairs of PL peak/ WL dip are clearly matching within their linewidth.

Figure R9: Photoluminescence (PL, blue) and white light (WL, red) reflectivity spectrum of a mixed phase layered perovskite flake derived from the same bulk crystal used for the polariton condensation experiments. Two resonances corresponding to the $n=3$ and the $n=4$ exciton are clearly visible in both PL and WL reflectivity.

2. Second, we investigated the strong coupling behavior of this flake within the 1.78 to 2.05 eV range. The data is shown as Fig. R10 below and now included as Fig S10 in the revised Supplementary information. At 2 eV, a Rabi splitting associated with the $n=3$ phase HaP exciton is observed, with a coupling strength of 16 meV. The reduced coupling strength compared to the data in Fig. 1 of the main text is attributed to the reduced flake thickness of 200 nm. Importantly, no evidence of a Rabi gap is observed at 1.92 eV, the energy of the $n=4$ phase HaP exciton. Instead, the data reveals a clear dispersive behavior attributed to the variation in the real part of the refractive index. This is in agreement with the much lower absorption for $n=4$ shown in Fig. R9. We note that the dispersive feature is almost absent in the device discussed in the main text, which hints at an even lower contribution from the $n=4$ phase. These results lead to the conclusion that the $n=4$ HaP phase does not couple strongly to the cavity photon and can hence act as an effective intra-cavity pump as discussed in the main text and supported by the absence of strong coupling features around 1.92 eV in the PL dispersion shown in Fig. S3.

Figure R10: Strong coupling measurement of a layered perovskite flake with a thickness of 200 nm. At 2 eV, the $n=3$ exciton shows strong coupling with the cavity photon, the $n=4$ exciton at 1.92eV only results in a dispersive feature and no Rabi gap. The dispersive feature is almost absent in the device discussed in the main text, for which full strong coupling data are shown in Fig. S2. This hints at an even lower $n=4$ contribution for the device used for polariton condensation experiments and is in full agreement with the absence of a Rabi gap at 1.92eV in the PL data presented in Fig. S3 of the Supplementary Information.

The reference in the main text to the new data in Fig. R10 reads:

“We observe clearly-developed anti-crossings for subsequent longitudinal mode orders (see Supplementary Sections S2 for a full range scan and S10 for additional strong coupling data on another flake from the same bulk crystal).”

3. Fig. R11 shows a microscope image of the fully hBN-encapsulated perovskite flake under white light illumination. The homogeneous dark red part is the perovskite while the other portions appearing light grey are hBN. This image further corroborates that our perovskites are spatially homogenous.

Figure R11: New Supplementary Figure S1. Optical microscope image of the fully encapsulated quasi-2D HaP flake, deposited on the bottom DBR of the open microcavity. Dark red parts are the perovskite, grey parts are the encapsulating hBN flakes.

6) *Fig. 2d does not show a reduction in the FWHM. This graph does not make sense.*

To address this issue **we include a fully re-measured new dataset as Fig. 2** of the manuscript that illustrates the collapse more clearly. As stated in our reply to the reviewer's main point ii), we also show a reduction of the linewidth by measuring the increase of the temporal coherence.

7) *Why is the fringe pattern visible even below the threshold? How homogeneous is the sample?*

The confined optical mode of the trap itself gives rise to a finite coherence length associated with the optical linewidth. As we raise the excitation power above threshold, this coherence length increases.

The homogeneity of the sample is now clearly underpinned by measuring near-identical polariton lasing behavior at different spatial positions on the same sample (new Fig. 2, main text and new Fig. S6, supplementary).

We thank the reviewer again for the critical assessment of our work. We trust that, based on our revisions and additions, the reviewer will agree with us that the demonstration of polariton condensation is unambiguous in our experiments and that our manuscript can be accepted for publication in Nature Communications.

Reviewer #3 (Remarks to the Author):

In the manuscript titled "Room-temperature polariton condensate in a two-dimensional hybrid perovskite," Struve et al. report the observation of polariton lasing at room temperature from layered perovskite materials embedded in a vertical microcavity. By monitoring the reflectivity spectra while scanning the exciton-photon detuning, the authors demonstrate strong coupling with evidence of the avoided-crossing effect between perovskite excitons and the photonic mode. Lasing emission was observed when the planar cavity was replaced by a 0D cavity confinement, induced by sphere-cap-shaped structures in the top mirror. The authors analyzed the nonlinearity (via energy blueshift) and spatial coherence (via interferometry measurement) of the lasing emission. Based on their measurements, the effects are attributed to polariton condensation.

While the results are scientifically sound, I have several concerns with this work that prevent me from recommending it for publication:

We thank the reviewer for the careful assessment of our manuscript. We are happy that the reviewer is convinced that the results are scientifically sound. Below, we address each of the reviewer's criticism in detail.

1. The reported lasing effect is photon lasing, as the excitonic fraction from the polaritonic mode is extremely small. Indeed, the excitonic fraction is about 1.3% if we use the light-matter coupling strength of $g = 23.5$ meV, an exciton energy of 2 eV, and a photon energy of 1.8 eV. Thus, the lasing emission is 98.7% photonic-like if attributed to polariton lasing. We should not expect any interesting polaritonic features from such a low excitonic fraction.

We understand the concerns regarding the low excitonic fraction for a strongly red detuned polariton. However, our results clearly retain polariton fingerprints. Based on the measured blueshift, we have compelling evidence supporting polaritonic behavior.

From our experimental experience it is crucial to choose the energy of the polariton below the $n=4$ absorption peak. In this case, the weakly coupled $n=4$ transition acts as an intra-cavity pump to further populate the polariton states. If the polariton energy is set higher, the absorption of the $n=4$ overpowers the photoluminescence, leading to a diminished signal. Therefore, we have to choose the energy of the polariton carefully to obtain a large signal while maintaining a meaningful excitonic fraction.

To address the reviewer's point, we conducted a second measurement on the same sample, but with a slightly increased exciton fraction. This measurement is included in the revised manuscript as new Fig. 2 (Fig. R12 below) while the old measurements are transferred to the Supplementary Information as Fig. S6. We have revised the discussion around it accordingly.

In the new measurement, the polariton energy was chosen to be 1.81 eV, leading to slightly higher Hopfield coefficients (1.5 % exciton fraction). These higher Hopfield coefficients result in a blueshift that is twice as large, further reinforcing the presence of distinct excitonic contributions.

The new Fig. 2 is included below as Fig. R12. We trust that the reviewer agrees that the polaritonic fingerprints (in particular the polariton blueshift) have become obvious.

Figure R12: New Figure 2 of the main text with fully re-measured input-output characteristic on a different area of the same quasi-2D HaP flake. The blueshift across the polariton condensation threshold is twice as large as before and the reduction of polariton linewidth at the threshold is clearly visible. We attribute both to a larger exciton fraction of the polariton leading to enhanced interaction-driven non-linearities.

Please note that we also enhanced the accuracy of our fits by using Voigt profiles. This is described in a new part of the methods section, which reads:

“We extract the input-output characteristic, linewidth and blueshift in Fig. 2b-d by fitting Voigt profiles to the output spectra in Fig. 2a. We approximate this profile as the weighted linear combination of a Lorentzian and a Gaussian, sharing the same center energy and linewidth. We consistently find that an area weight of 0.69 ± 0.03 for the Lorentzian results in the best approximation to the experimentally observed lines.”

2. The nonlinearity investigation is based on an energy blueshift ranging from 0 to 0.36 meV (Fig. 2c), while the polariton linewidth is about 1.6 meV, several times larger. Therefore, the nonlinearity presented here is not significant. Moreover, the blueshift in perovskite material emission could also be attributed to thermal effects.

While it is true that the observed blueshift of our polariton system is relatively small compared to its linewidth, the shift lies within our experimental accuracy and is therefore significant. The blueshift and linewidth are independent from each other and should not be seen in relation to each other. At the same time, our theoretical calculations indicate that the expected blueshift is inherently small. Interestingly, the experimentally measured blueshift exceeds our model, most likely due to the presence of a dark exciton population leading to phase-space filling effects.

Additionally, in our newly acquired dataset (Fig. R12, i.e. the new Fig. 2 of the main text), we observe a blueshift of 0.8 meV, that is approximately twice as large as the previously measured shift.

Furthermore, we conducted temperature-dependent photoluminescence measurements (Fig. R13) which reveal that heating of the n=3 perovskite from room temperature results in a pronounced **redshift** of the PL, with shifts reaching up to 14 meV for an increase in temperature by 40 °C. **This finding rules out thermal effects as the origin of the observed blueshift** in our system.

We now include Fig. R13 in the Supplementary as Fig. S4 and describe it in the main text.

Figure R13: Temperature-dependent PL of an n=3 phase quasi-2D HaP flake deposited on a DBR mirror. The mirror was heated from the back side with a Peltier element, while PL was measured under non-resonant CW excitation at 532nm. Heating by 40 K from room temperature results in a large red shift by 14meV. This measurement rules out thermal effects as the origin of the observed blueshift in our system. We find that the red shift was reversed after cooling down the sample.

The added description in the main text reads:

“To rule out thermal effects as the origin of the observed blueshift, we measured the temperature dependence of the quasi-2D HaP PL, observing a pronounced thermal redshift (see Supplementary Section S6).”

3. Regarding the coherence measurement, the same effect is expected from pure photon lasing. Since the lasing emission originates from a well-defined photonic OD mode, spatial coherence is naturally expected within the spatial profile of this mode (which could extend beyond the shape of the sphere cap due to shallow confinement).

We certainly agree. The emergence of coherence is a general benchmark of lasing phenomena and occurs in both polariton and photon lasing. The distinction between polariton lasing and photon lasing is demonstrated through the blueshift of the polariton condensate in our work. The measurement of the coherence at this point not only demonstrates the existence of coherence but importantly also the rise of coherence above the threshold.

Additionally, I have concerns regarding the novelty of this work, which I do not find suitable for Nature Communications. Room-temperature polariton lasing has already been reported in perovskites by other groups, with even stronger nonlinear effects such as optical parametric oscillation and superfluidity. In terms of materials, previous reports used 3D inorganic perovskites and 3D hybrid perovskites, and I do not see how using layered perovskites introduces novelty.

We appreciate the reviewer's feedback. While room-temperature polariton lasing has indeed been previously reported in 3D bulk perovskites, our work highlights significant advantages of 2D layered perovskites over their 3D counterparts, which adds substantial novelty to our work.

• We now extended and clarified the discussion on novelty both in the introduction and conclusion of our revised manuscript based on the following arguments:

1. Unlike for their 3D counterparts, choosing the thickness of the inorganic layers in terms of the number of unit cells, n , results in a unique control over the exciton resonances in quasi-2D HaPs, resulting in flexible spectral emission and absorption characteristics for optoelectronic devices.
2. In quasi-2D HaPs, the exciton binding energy is a separate, directly accessible tuning parameter, e.g. via doping the organic spacer. This has no analog in 3D HaPs.
3. 2D HaPs can be micromechanically cleaved and re-assembled in a well-defined manner. This has already lead to first demonstrations of charge-tunable gated devices in 2D HaPs with electrically controlled optical response and clearly hints at the possibility of realizing electrically driven room-temperature polariton lasers based on quasi-2D HaPs with potentially unprecedented flexibility in terms of emission wavelength and in-situ tenability.
4. Being van-der-Waals materials, quasi-2D HaPs stand out against their bulk counterparts, since quasi-2D HaPs offer the benefit of easy hetero-integration typically found in TMDCs. This opens an exciting new field of research for room-temperature polariton condensates based on hybrid heterostructures, integrated with electrically tunable cavities or subjected to tunable photonic lattices.

The corresponding revised section in the introduction reads:

"[...] Unlike for their 3D counterparts, which have recently emerged as a promising material platform for monolithic room-temperature polariton devices [16-20], choosing the thickness of the inorganic layers in terms of the number of unit cells, n , results in a unique control over the exciton resonances in quasi-2D HaPs [7,21-23]. Even the exciton binding energy is a separate, directly accessible tuning parameter in quasi-2D HaPs, e.g. via doping the organic spacer [24]. Most importantly, 2D HaPs can be micromechanically cleaved and re-assembled in a well-defined manner [4,25-28]. This has already lead to first demonstrations of charge-tunable gated devices in 2D HaPs [29] with electrically controlled optical response.

The giant excitonic oscillator strengths and binding energies, alongside with spectral tunability and versatility for complex hetero-integration render quasi-2D HaPs a particularly promising material class for exploring room temperature polaritonics. The formation of cavity exciton-polaritons based on these materials has recently been observed in planar dielectric, metallic and hybrid cavities [30-36]. However, in quasi-2D HaPs, the bosonic condensation of exciton-polaritons [37,38] - their most significant feature for applications based on their quantum coherence - has only been demonstrated at 4 Kelvin in the 2D HaP $(\text{PEA})_2\text{PbI}_4$ ($n = 1$, PEA=phenethylammonium) [39], and so far remained elusive at room temperature. [...]"

The corresponding section in the outlook reads:

“Quasi-2D halide perovskites stand out in the landscape of polariton material platforms due to an unprecedented combination of key advantages: Being chemically synthesized multi-quantum well stacks, they retain the unique flexibility in emission wavelength and oscillator strength of conventional QW stacks via controlling the thickness and number of inorganic layers. Choosing and doping the inorganic spacers even controls the exciton binding energy as a separate parameter [24]. This set of advantages is paired with excellent room-temperature operability also found in their bulk HaP counterparts, which have recently emerged as a viable platform for monolithic room-temperature polariton devices [16-20]. However, being van-der-Waals materials, quasi-2D HaPs stand out against their bulk counterparts, since quasi-2D HaPs offer the benefit of easy hetero-integration typically found in TMDCs. This opens an exciting new field of research for room-temperature polariton condensates based on hybrid heterostructures [28], integrated with electrically tunable cavities [55] or subjected to tunable photonic lattices [41,43,56]. First demonstrations of charge tunable devices in quasi-2D HaPs [29] clearly hint at the possibility of implementing electrically driven polariton lasers based on quasi-2D HaPs with potentially unprecedented flexibility in terms of emission wavelength and in-situ tunability. Based on the fast-paced progress with HaPs in the fields of photovoltaics and optoelectronics, we anticipate that the implementation of electrically driven room-temperature polariton devices using quasi-2D HaPs is realistic in the near future resulting in vast opportunities for on-chip integration.”

Other groups have reported polariton lasing with Rabi splittings in the range of hundreds of meV, an order of magnitude higher than what is reported here. This indicates that the perovskite material used in this work has much weaker oscillator strength.

The main reason for the smaller Rabi splitting in comparison to results from other groups is the thickness of the perovskite material that we used. In layered perovskites, the thickness of the material corresponds directly to the number of quantum wells and therefore to the coupling strength. In many studies, materials of several micrometers are grown on a substrate, while we mechanically exfoliate a crystal of 200 nm thickness.

We, however, emphasize that our transfer matrix simulations (Fig. S2) are purely based on the measured refractive indices of $n=3$ perovskite at room temperature [Song et. al. *ACS Mater. Lett.* **3**, 148–159 (2021)] in conjunction with AFM determination of flake thickness. These simulations demonstrate perfect agreement with experimentally measured values for the Rabi splitting. Therefore, we can state with high confidence that a Rabi splitting of 23 meV is to be expected for our material.

We now emphasize this connection more clearly in the main text:

“At an air gap of $3.3\ \mu\text{m}$, we extract a light-matter coupling strength of $g = 23.5\ \text{meV}$ based on a coupled oscillator model (dashed lines in panel c). The magnitude of the Rabi gap is in excellent agreement with transfer matrix simulations based on dielectric functions for the quasi-2D HaP taken from Ref. [8] (see Supplementary Sections S2 and S3 for details).”

Additional comments:

4. *The authors refer to their perovskite as “2D perovskite,” which is incorrect. It should be termed “layered perovskite” since $n = 3$, not the 2D perovskite case where $n = 1$.*

We followed the suggestion by the reviewer and changed the nomenclature to “quasi-2D” in the manuscript.

This nomenclature is widely accepted in the community, e.g. [Nature Materials volume 23, pages 182–188 (2024), Nature Communications volume 12, Article number: 2207 (2021) Light: Science & Applications volume 10, Article number: 61 (2021), Energy Environ. Sci., 2022,15, 2499-2507].

5. *I am curious why the Rabi splitting is so small in this work, as typical Rabi splitting in perovskite materials (2D, 3D, hybrid, all-inorganic) is around 100-200 meV.*

The smaller Rabi splitting is caused by the much smaller amount of material in our case, i.e. quasi-2D Perovskite flakes of 200 nm thickness. We however emphasize that our observation of a Rabi splitting of 23 meV is in complete agreement with the underlying dielectric function of our material [Song et. al. *ACS Mater. Lett.* **3**, 148–159 (2021)].

In other samples, which were based on thicker layers, and which were not optimized for polariton lasing, we indeed reached Rabi splittings comparable to the exciton binding energy (100 meV).

We thank the reviewer again for the pertinent comments to our work and hope that the revised manuscript can be accepted for publication in Nature Communications.

Blue italics: Reviewer comment

Black: Our response

Red: Changes to the manuscript/SI

Reviewer #1 (Remarks to the Author):

I believe the authors have adequately addressed my previous comments in the revised manuscript. Based on the improvements made, I am pleased to recommend its publication in Nature Communications.

We thank the reviewer for their time and effort to review our work. We are happy that he/she recommends it for publication in Nature Communications.

Reviewer #2 (Remarks to the Author):

I appreciate the authors for addressing my concerns, providing additional experimental results, and offering further interpretations of the data. The revised version of the manuscript is notably improved and provides a clearer explanation of the measured effect.

We are pleased that the reviewer finds our revised version has notably improved. Below, we respond in detail to each of the reviewer's remaining questions. In particular, we have performed extensive additional experiments to address the concerns regarding the role of the n=4 phase and the lateral confinement.

I still have a couple of concerns.

1) Can the authors experimentally demonstrate the role of the n=4 phase in polariton lasing? It would also be useful to include a comparison with a sample where the n=4 phase is absent.

To clarify the role of the n=4 phase in our experiments, we followed the reviewer's suggestion and **ran extensive additional polariton lasing studies** on a pure n=3 phase perovskite crystal, which we now include in the Supplementary Information. The results are shown below in Figs. R1 and R2. Fig. R1 shows white light reflectivity (WL) and photoluminescence (PL) spectra of a pure n=3 phase $(BA)_2(MA)_2Pb_3I_{10}$ flake that was prepared following the same procedure as outlined in the methods section of our manuscript. Comparing these spectra to Fig. S9 of our supplementary material (PL and WL for mixed phase crystal), the n=3 contributions are clearly visible in WL as an absorptive feature at 2eV and a single peak in PL at roughly the same energy. The features attributed to n=4 (shallow absorption in WL and second peak in PL at 1.93eV) are absent.

Figure R1: Photoluminescence (PL, black) and white light (WL, red) reflectivity spectrum of a pure n=3 phase layered perovskite flake derived from the same bulk crystal used for the polariton condensation experiments. The resonance corresponding to the n=3 exciton is clearly visible in both PL and WL reflectivity.

Figure R2: Input-output curve acquired on the pure $n=3$ sample characterized in Fig. R1. Double logarithmic plot of the input-output curve extracted from the areas under Voigt fits to the emission spectra (analogous procedure to the data analysis for the mixed phase crystal presented in Fig. 2 of the main text). The same excitation conditions and the same sphere-cap upper mirror of the cavity were used. For the same range of input fluences as shown in Fig. 2 no sign of an optical non-linearity is observed.

We performed polariton lasing experiments on this new sample following the exact same procedure used for the mixed phase perovskite in Fig. 2 of the main text. The flake was fully hBN encapsulated and placed on a DBR mirror equal to the ones used in the main text. We completed the open cavity with the same sphere cap shaped indentation used before.

Input-fluence dependent PL spectra were evaluated in the same way as for Fig. 2 of the main text. The resulting input-output curve showing the area under a Voigt-fit to the PL spectrum as a function of input fluence in double logarithmic representation is shown in Fig. R2. We observe a linear behavior with no sign of optical non-linearity.

From these results we conclude that the presence of the $n=4$ contribution in our mixed phase crystals is instrumental in observing polariton lasing in our case. The most likely scenario for the origin of this effect is the intra-cavity pumping outlined in our manuscript.

To maximize the likelihood of polariton lasing to occur for the pure $n=3$ phase, we tuned the localized polariton mode to an energy slightly below 1.9eV, i.e. blue-shifted from the energy chosen for the mixed phase. This choice of energy on the flank of the PL emission spectrum of the pure $n=3$ phase but still in the high reflectivity range of the WL reflectivity spectrum in Fig. R1 maximizes PL emission under non-resonant CW excitation and is the same strategy that we used to choose the optimum detuning in the mixed phase case. We also tested the $n=3$ phase crystal under the same detuning as chosen for Fig. 2 of the main text, yet PL intensity was too low to result in a meaningful input-output characterization.

Changes to the manuscript/SI:

Figures R1 and R2 are now included in the SI as new Figures S11 and S12 in the new SI section S11, which describes our findings from the response above. Section S11 is now referenced in the main text on page 4:

“Under otherwise equal conditions, the optical non-linearity is not observed for pure $n=3$ phase quasi-2D HaP (see Supplementary Section S11), underpinning the importance of the weakly coupled $n=4$ phase in our case.”

2) What is the physical reason for the weak coupling between the n=4 phase and the optical mode?

To further clarify why the n=4 part of the mixed-phase quasi-2D perovskite couples weakly to the optical cavity mode – in contrast to the n=3 part – we performed a set of transfer matrix simulations to analyze this effect. In these calculations, we simulated white light reflectivity spectra of the planar cavity configuration as a function of cavity distance. We replaced the n=3 pure phase perovskite by an effective medium, which we implemented as a weighted average of the refractive indices for n=3 and n=4. To illustrate the effect in principle, we chose equal ratios of the two components. The results of the calculation are shown in Fig. R3 below. We observe clear anti-crossings between the n=3 exciton resonance and subsequent longitudinal cavity photon modes slightly above 2eV. However, at the n=4 exciton resonance around 1.92eV, no Rabi gap is present, indicative of the weak coupling between the n=4 exciton and the cavity modes. **This calculation is in very good principal agreement with the corresponding experimental data in Fig. S10.**

Figure R3: Transfer matrix simulation of white light reflectivity spectra as a function of the cavity air gap at normal incidence ($k_{\parallel} = 0$). The active medium is an effective medium obtained as a weighted average of the refractive indices for n=3 and n=4 with equal ratios. While a Rabi gap is visible for the n=3 exciton resonance slightly above 2eV, there is no Rabi gap present at the position of the n=4 exciton at 1.92eV.

The reason why the n=4 phase couples weakly in the present scenario, becomes evident: While pure phase n=3, as well as pure phase n=4 perovskite both display the Rabi-gap in the simulation, the n=4 exciton loses oscillator strength via the averaging the two refractive indices which results in the transition to the weak coupling regime. Since the n=3 exciton transition has a higher energy, averaging with the refractive index of n=4 results in persistent contributions of the n=4 Lorentz oscillator to the oscillator strength of the n=3 resonance. These contributions are strong enough to retain the strong coupling conditions for n=3.

Changes to the manuscript/SI:

We include the main arguments of our response above in Supplementary Section S10 describing the experimental observation of weak coupling for the n=4 phase. The section is already appropriately referenced in the main text on page 4.

3) The actual effect of the OD confinement induced by the sphere on the DBR is not entirely clear. Does this effect also occur in the planar section? How does it impact the polariton threshold—does it change or disappear?

We thank the reviewer for raising this important point. To answer the question whether the OD confinement is instrumental in reaching the polariton condensation regime in our system, **we have performed extensive additional input-output studies on planar sections of our microcavity**. The results of these measurements are shown in Fig. R4 and are now included in new Supplementary Section S12. Figure R4 (a) and (b) show momentum-resolved PL under pulsed non-resonant excitation for the same mixed phase perovskite flake as used in the dataset for Fig. S9/10 of the supplementary information. The detuning and upper mirror were the same as in the main text, but we used a planar section of the upper mirror. For fluences slightly above (a) and well above (b) the polariton condensation threshold reported in the main text, no distinct change in PL distribution towards a macroscopic occupation around $k=0$ is observed.

Figure R3: a,b: momentum-resolved PL under pulsed non-resonant excitation for the same mixed phase perovskite flake as used in the dataset for Fig. S9/10 of the supplementary information. The detuning and upper mirror were the same as in the main text, but we used a planar section of the upper mirror. For fluences slightly (a) and well above (b) the polariton condensation threshold reported in the main text, no distinct change in PL distribution towards a macroscopic occupation around $k=0$ is observed. c,d: Input-output curve acquired on the sample

We furthermore recorded detailed input-output characteristics of the planar cavity device. The results are presented in panel (c) showing a fine scan for small pump fluences in the fluence range for which the polariton condensation threshold was observed with OD confinement and in panel (d) showing a coarser scan over a wider range of pump fluences reaching up to $35 \cdot P_{th}$ with $P_{th} = 0.41 \mu J/cm^2$ the threshold observed with OD confinement under otherwise equal conditions on the same perovskite flake. Both input-output curves in panels (c) and (d) show no deviation from a linear trend

underpinning that OD confinement is indeed largely helpful for reaching polariton condensation at very moderate pump fluences.

We note that the enhancement of stimulated scattering and the trend towards lower polariton lasing thresholds under OD confinement is well established in the literature. The enhancement of stimulated scattering in confined polariton system was discussed in detail in Paraiso et al. Phys. Rev. B 79 045319 (2009). D. Bajoni et al. Phys. Rev. Lett. **100**, 047401 (2008) and A. Askitopolous et al Phys. Rev. B. 88 041308 (2013) for example exploited the phenomenon for demonstrations of polariton lasing in monolithic III-V semiconductor microcavities where OD confinement was established by laterally etching the microcavity into micropillars or by defining an optical trap.

Changes to the manuscript/SI:

Figure R3 and the surrounding text are now included as a new Supplementary Section S12 and Supplementary Figure S13 in the Supplementary information.

We summarize our response in the main text. We included the following sentence on page 4

“The confinement is instrumental in reaching polariton condensation in our system, since it enhances stimulated scattering⁴⁷ (see Supplementary Section S12 for direct comparison to a planar cavity). This strategy has been exploited before, for example for demonstrations of polariton lasing in monolithic III-V semiconductor cavities under transverse confinement from micropillars⁴⁸.”

We thank the reviewer again for their time and the pertinent comments. We hope that with the added information in place, our work can now be published in Nature Communications.

Reviewer #3 (Remarks to the Author):

The authors have addressed most of my questions from the previous report and have provided substantially improved data along with additional experiments. I now believe that the reported results do correspond to polariton condensation. However, I would recommend publication of this work in Nature Communications only after another revision, as I still have several concerns regarding the new data set:

We are very pleased that the reviewer finds the revised version of our manuscript and the experimental data have much improved from the initial version. Below, we respond to each of the remaining questions raised by the reviewer.

1. The authors present new measurements with an excitonic fraction of 1.5%, compared to 1.3% previously—an increase of about 1.15 times. How can such a relatively small change result in a twofold increase in the blueshift?

We agree with the reviewer that this point requires an extended explanation, and we provide more details in the revised version (also summarized below). In the simplest case of exciton-exciton scattering the expected polariton blueshift should scale quadratically with the Hopfield coefficient (exciton fraction), $\Delta \sim |X|^4$. Hence, for otherwise the same conditions the observed variation should be 1.33, and below the two-fold increase. However, we note that there are several contributions that need to be accounted for when comparing results.

First, nonlinear shifts originate from the combination of exciton-exciton scattering and nonlinear phase space filling (saturation), slightly modifying the scaling, where the saturation-based shift scales as $|X|^{3/2}$ [Betzold et al., ACS Photonics 7, 384 (2020)]. Second, the absolute value of the nonlinear shift depends on the effective area A_{eff} that we measure, as well as the structural composition. The two sets of measurements, in Fig. 2 of the main text and in Fig. S6 of the Supplementary, were performed on the same perovskite flake but not at the exact same location. Although the PL spectra below polariton condensation threshold are nearly identical at these two locations, it is likely that the vertical composition of the mixed phase crystal showed slight variations leading to the observed changes (for instance, relative fraction of $n=3$ and $n=4$ perovskite phases). This could contribute to the modification of the effective Bohr radius. Also, since the disorder landscape is not completely identical, we could expect variations of A_{eff} at the level of tens of percent.

Finally, we note that the derived theoretical value agrees with the polariton blueshift observed in both measurements within approximately an order of magnitude (p. 5 in the main text and Supplementary Section S7). Here, we aim at the qualitative description of nonlinearity, and note that deviations allow for additional contributions (for instance, induced by residual dark exciton population). This level of agreement is rather common, even when one compares exciton-polariton nonlinearity in III-V materials [Estrecho et al., Phys. Rev. B 100, 035306 (2019)]. We clearly state in the revision that the study provides an indicative range of nonlinear scattering coefficients, rather than their exact deduction from measurements.

Changes to the manuscript/SI:

We clearly state on page 5 of the main text that the study provides an indicative range of nonlinear scattering coefficients, rather than their exact deduction from measurements:

“Our aim here is to provide an indicative range of nonlinear scattering coefficients by an order of magnitude comparison between reasonable theoretical estimates and the experiment.”

We added the detailed discussion provided above to Supplementary Section S8. On page 5 of the main text, we more clearly mention that the two datasets in the manuscript and supplementary show deviations in nonlinear scattering coefficients beyond the expectation from changes in Hopfield coefficients and reference the in-depth discussion in Supplementary Section S8:

“We also find that the measured non-linear scattering coefficient consistently remains within the same order of magnitude when comparing different places on the same flake. Observed variations in the coefficient are accounted for by its dependence on Hopfield coefficients and by slight modifications in measured effective area as well as structural composition of the flake (see Supplementary Section S7-S8 for details).”

The stated experimental values for the nonlinear interaction coefficients in Fig. 2d and S6 contained a typo. We corrected the values to $\approx 1.9 \mu\text{eV}\mu\text{m}^2$ (S6) and $\approx 5.4 \mu\text{eV}\mu\text{m}^2$ (2d) throughout the manuscript and SI.

2. Figure 2d, which shows the linewidth narrowing, is not discussed in the manuscript. In the case of polariton condensation in a planar cavity, linewidth narrowing above threshold is attributed to macroscopic occupation at $k = 0$, with negligible population at other momenta in the dispersive cavity. In contrast, for photon lasing in a trapped (i.e., nondispersive, discrete) state, narrowing is due to the dominance of stimulated emission in the lasing mode over spontaneous emission into other channels. However, the physical origin of linewidth narrowing in the case of polariton condensation in a trapped state remains unclear to me. Can the authors clarify this point?

We thank the reviewer for raising this important question. In a 0D photonic system (such as a VCSEL laser), the linewidth narrowing and associated rise in temporal coherence is due to the dominance of stimulated emission in the lasing mode over spontaneous emission, which requires population inversion. In analogy to stimulated emission, in a 0D polaritonic system the effect that becomes dominant above the lasing threshold is stimulated scattering into the polariton ground state.

This important distinction and the fact that bosonic final state stimulation is responsible for the observable increase in temporal coherence in a trapped polariton system when driven above threshold has been discussed, e.g., in [D. Bajoni et al. Phys. Rev. Lett. **100**, 047401 (2008); H. Deng et al. Rev. Mod. Phys. **82**, 1489 (2010); I. Carusotto et al. Rev. Mod. Phys. **85**, 299 (2013); B. Zhang et al. Light: Science & Applications **3**, e135 (2014)].

Changes to the manuscript/SI:

Based on our response above, we included the following sentences in the discussion of Fig. 2d on page 4 of the main text:

“The increase in temporal coherence and associated linewidth narrowing observed for trapped polariton systems driven above threshold, as shown in our case, is usually assigned to dominating bosonic final state stimulation, as has been discussed, e.g., in ^{48,50-52}.”

3. With the improved linewidth extraction, can the authors now quantify the contributions to the linewidth: how much arises from the photonic mode, how much from the excitonic component, and what is the contribution from inhomogeneous broadening of the exciton?

We thank the reviewer for this interesting suggestion. With the improved linewidth extraction, we can indeed quantify the relative contribution of Lorentzian and Gaussian components to the overall polariton linewidth. We find, as stated in the methods section of the main text (subsection Data analysis), that the overall lineshape is well accounted for by a linear superposition of 70% Lorentzian and 30% Gaussian with equal linewidth (Pseudo-Voigt fit).

There are, however, two possible origins for the presence of the Gaussian component:

- Dynamic broadening due to the open nature of our optical cavity. That is, the relative cavity mirror distance may fluctuate slightly at frequencies much higher than the detection bandwidth of our optical spectrometer.
- Dynamic broadening due to the ultrafast pulsed laser excitation with pulse duration well below the polariton lifetime.

Directly assigning the Gaussian broadening to one of the two or even disentangling their relative contribution is not directly possible.

In contrast, the Lorentzian linewidth of a cavity exciton polariton is given by the sum of the individual linewidth for the bare photonic cavity and the bare exciton, each weighted with their corresponding Hopfield coefficient. This is under the assumption that motional narrowing may be neglected. In this context, the discussion of our particular system, however, requires a little more care, since in the mixed phase perovskite case we also have contributions to the polariton linewidth of the weakly coupled $n=4$ component: We may effectively describe the resulting polariton linewidth below threshold as composed of the bare photonic cavity including the losses through absorption by the $n=4$ component weighted with the cavity Hopfield coefficient, and the exciton linewidth of the $n=3$ exciton with the exciton Hopfield coefficient as its weight. This means, for the configuration of the experiment in Fig. 2 of the main text, the main contribution to the observed polariton linewidth below threshold is the bare cavity including absorptive losses from the $n=4$ perovskite.

It has been shown before that inhomogeneous broadening of the exciton does not result in inhomogeneous broadening of a cavity exciton polariton, if the exciton enters the strong coupling regime. This effect is the central result of a series of seminal papers, including [R. Houdré et al. Phys. Rev. A 54 2711 (1996), C. Ell et al. Phys. Rev. Lett. 80, 4795 (1998)].

We thus conclude that it is unlikely that the observed Gaussian component to the polariton line is caused by inhomogeneous broadening of the $n=3$ exciton.

Changes to the manuscript/SI:

We include the main point of the above discussion in the main text on page 4:

“We assign the Gaussian contribution of the Voigt lineshape to a combination of dynamic broadening (via polariton blueshift) and dynamic fluctuations of the cavity length.”

We hope that with these added information and data included, our work can be published in Nature Communications. We thank the reviewer again for their helpful comments and questions.